

# Variability of phyto- and zooplankton communities

## in the Mauritanian coastal upwelling between 2003 and 2008

*Oscar E. Romero[1], Karl-Heinz Baumann[1,2], Karin A. F. Zonneveld[1], Barbara Donner[1], Jens Hefter[3], Bambaye Hamady[4] and Gerhard Fischer[1,2]*

[1]University of Bremen, Marum, Center for Marine Environmental Sciences, Leobener Str. 8, 28359 Bremen, Germany.

[2]University of Bremen, Department of Geosciences, Klagenfurter Str. 2-4, 28359 Bremen, Germany.

[3]Alfred Wegener Institute, Helmholtz Centre for Polar and Marine Research, 27568 Bremerhaven, Germany.

[4]IMROP, Institut Mauretanien de Recherches Océanographiques et des Pêches, BP 22, Nouadhibou, Mauritania.

**Keywords:** Eastern Boundary Upwelling Ecosystems, fluxes of microorganisms, interannual variability, northwest Africa, primary producers, secondary producers, sediment traps

## Abstract

Continuous multiyear records of sediment trap-gained microorganism fluxes are scarce. Such studies are important to identify and to understand the main forcings behind seasonal and multiannual evolution of microorganism flux dynamics. Here, we assess the long-term flux variations and population dynamics of diatoms, coccolithophores, calcareous and organic dinoflagellates, foraminifera and pteropods in the Eastern Boundary Upwelling Ecosystem (EBUE) of the Canary Current. A multiannual, continuous sediment trap experiment was conducted at the mooring site $CB_{eu}$ (*Cape Blanc eutrophic*, ~20°N, 18°W; trap depth = ca. 1,300 m) off Cape Blanc, Mauritania (northwest Africa), between June 2003 and March 2008. Throughout the study, the reasonably consistent good match of fluxes of microorganisms and bulk mass reflects the seasonal occurrence of the main upwelling season and the contribution of microorganisms to mass flux off Mauritania. A clear successional pattern of microorganisms, *i.e.* primary producers followed by secondary producers, is not observed. High fluxes of diatoms, coccolithophores, organic dinoflagellates cysts, and planktonic foraminifera occur simultaneously. Peaks of calcareous



dinoflagellate cysts and pteropods mostly occurred during intervals of upwelling relaxation. A

striking feature of the temporal variability of populations' occurrence is the persistent pattern of

32 seasonal groups' contribution. Species of planktonic foraminifera, diatom, and organic

dinoflagellate cysts typical of coastal upwelling as well as cooler water planktonic foraminifera and

34 the coccolithophore *Gephyrocapsa oceanica* are abundant at times of intense upwelling (late

winter through early summer). Planktonic foraminifera and calcareous dinoflagellate cysts

dominant in warm pelagic surface waters and all pteropod groups are more abundant in fall and

winter, when the mixed layer deepens. Similarly, coccolithophores of the upper- and lower photic

zone, together with *Emiliania huxleyi,* and organic dinoflagellate cysts dominate the assemblage

during phases of upwelling relaxation and deeper layer mixing.

A significant shift in the 'regular"' seasonal pattern of species relative contributions is observed

between 2004 and 2006. Benthic diatoms strongly increased after fall 2005 and dominated the

42 diatom assemblage during main upwelling season. Additional evidence for a change in population

dynamics are the short dominance of the coccolithophore *Umbilicosphaera annulus,* the

44 occurrence of the pteropod *Limacina bulimoides,* and the strong increase in the flux of calcareous

dinoflagellate cysts, abundant in tropical, warm oligotrophic waters south of the research area after

46 fall 2005. Altogether, this suggests that pulses of southern waters were transported to the sampling

site via the northward Mauritania Current. Our multiannual trap experiment provides a unique

opportunity to characterize temporal patterns of variability that can be extrapolated to other

EBUEs, which are experiencing or might experience similar future changes in the plankton

community.

**1. Introduction**

A way to obtain insights into the impact of climate variability on marine ecosystems is

54 monitoring multiannual evolution and changes covering key species or groups of species

representing different trophic levels. To date, continuous *in situ* long-term, monitoring records of

56 marine communities are scarce. Information about open oceanic ecosystems is even more rare

(see *e.g.* overview of currently available long-term time series of phytoplankton community



abundance and composition IOC-UNESCO TS129 IGMETS 2017). Furthermore, records providing

information about organism groups of different trophic levels are practically unknown or cover only

a few species (*e.g.*, Schlüter et al., 2012; Rembauville et al., 2016).

Eastern Boundary Upwelling Ecosystems (EBUEs) are among the most important marine

ecosystems, both ecologically and economically (Cropper et al., 2014). Despite the fact that they

cover only 10% of the global surface ocean area, they provide about 25% of the global fish catch

(Pauly and Christensen, 1995) and build extraordinary marine hotspots of high primary production

and biodiversity (Arístegui et al., 2009). In doing so, they play a key role in the marine biological

pump, as together with other continental margins may be responsible for more than 40% of the

$CO_2$ ocean sequestration (Muller-Karger et al., 2005). As EBUEs are highly dynamic with large

seasonal and interannual variability (*e.g.*, Chavez and Messié, 2009; Fischer et al., 2016), gaining

information on their long-term variability is essential to understand their potential response to

current global climate change.

One of the EBUEs that have been thoroughly studied over the past three decades is the coastal

ecosystem off Mauritania (northwest Africa), which is part of the Canary Current (CC) EBUE

(Cropper et al., 2014). The Mauritanian system is characterized by intense offshore Ekman

transport and strong mesoscale heterogeneity, which facilitate the exchange of neritic and pelagic

water masses (Mittelstaedt, 1983; Zenk et al., 1991; Van Camp et al., 1991; Arístegui et al., 2009;

Chavez and Messié, 2009; Meunier et al., 2012; Cropper et al., 2014). In addition, regional factors

such as nutrient trapping efficiency (Arístegui et al., 2009), the giant chlorophyll filament (Gabric et

al., 1993; Barton, 1998; Lange et al., 1998; Helmke et al., 2005); dust deposition (Fischer et al.,

2016, 2019) and/or the shelf width (Hagen, 2001; Cropper et al., 2014) strongly affect the temporal

dynamics of primary and secondary producers communities in surface waters along the

Mauritanian coast. In this ecosystem, several long-term, continuous, sediment trap-based

monitoring records are available since the late 1980s. Until now, studies monitoring variability of

this seasonally dynamic ecosystem mostly focused on the variability of bulk fluxes (Fischer et al.,

1996, 2009, 2016, 2019; Bory et al., 2001; Marcello et al., 2011; Skonieczy et al., 2013), particular

groups of microorganisms (Lange et al. 1998; Romero et al., 1999, 2002, 2003; Köbrich and

Baumann, 2008; Romero and Armand, 2010; Zonneveld et al., 2010; Köbrich et al., 2016; Romero



and Fischer, 2017; Guerreiro et al., 2019) or sea-surface temperature (Müller and Fischer, 2001;

Mollenhauer et al., 2015). However, the simultaneous comparison of the seasonal and interannual

dynamics of several phyto- and zooplankton communities by means of multiyear sediment trap

experiments has not been performed in this region and is rare in other EBUEs or other ocean

areas as well.

In this study, we describe the seasonal and interannual variability of fluxes of several primary

and secondary producers in the Mauritanian coastal upwelling over a continuous trap experiment

extending almost 1,900 days between June 2003 and March 2008 (Table 1). We present fluxes of

diatoms, coccolithophores, calcareous and organic-walled dinoflagellate cysts, planktonic

foraminifera and pteropods as well as the species-specific composition of the assemblages that

have been collected at the mooring site CBeu (*Cape Blanc eutrophic*), located around 80 nm west

of the Mauritanian coastline below a giant chlorophyll filament (Fig. 1). The organisms producing

the calcareous, opaline or organic remains have different water column habitats, life strategies,

and use different nutrient sources (Romero et al., 1999, 2002; Baumann et al., 2005; Romero and

Armand, 2010; Zonneveld et al., 2013), and are widely used as proxies in paleostudies carried on

Mauritanian sediments (Jordan et al., 1996; Romero et al., 2008; McKay et al., 2014) and similar

paleoenvironments (*e.g.*, Baumann and Freitag, 2004; Bouimetarhan et al., 2009; Romero et al.,

2015; Weiser et al., 2016; Hardy et al., 2018). The emphasis of our multiannual trap experiment is

on the comparison of temporal dynamics and the species-specific composition of the primary and

secondary producer plankton community off Mauritania. The simultaneous assessment of fluxes of

several microorganism groups collected over long intervals provides substantial information about

potential changes in the coastal upwelling community. Results are discussed in the context of

varying environmental conditions of the low-latitude Northeastern Atlantic. To our knowledge, this

is the first multiyear trap-based record of primary and secondary producers that provides multiyear

information on the dynamics of populations in a highly productive coastal upwelling system.





**2. Oceanographic setting of the study area**

The CC-EBUE is located in the eastern part of the North Atlantic Subtropical Gyre (Fig. 1;

Arístegui et al., 2009; Chavez and Messié, 2009; Cropper et al., 2014). Both the temporal

occurrence and the intensity of the upwelling along northwestern Africa depend on the shelf width,

the seafloor topography, and wind direction and strength (Mittelstaedt, 1983; Hagen, 2001). The

Mauritanian shelf is wider than the shelf northward and southward and gently slopes from the

120 coastline into water depths below 200 m (Fig. 1b; Hagen, 2001). The shelf break zone with its

steep continental slope extends over a distance of approximately 100 km (Hagen, 2001). As a

122 result of the coastal and bottom topography, and the ocean currents and wind systems, the coastal

region off Mauritania is characterized by almost permanent upwelling. Its intensity varies

throughout the year (Lathuilière et al., 2008; Cropper et al., 2014). Our trap site $CB_{eu}$ is located at

the southern boundary of this permanent upwelling zone (Fig. 1; Table 1).

The surface hydrography is strongly influenced by two wind-driven surface currents: the

southwestward-flowing CC and the poleward-flowing coastal countercurrent or Mauritania Current

(MC) (Fig. 1). The eastern branch of the subtropical gyre, the surficial CC detaches from the

continental slope between 25° and 21°N and supplies waters to the North Equatorial Current. The

130 CC water is relatively cool because it entrains upwelled water from the coast as it moves

southward (Mittelstaedt, 1991). The MC gradually flows northward along the coast up to about

132 20°N (Mittelstaedt, 1991), bringing warmer surface waters from the equatorial realm into the study

area. Towards late autumn, the MC is gradually replaced by a southward flow associated with

134 upwelling water due to the increasing influence of trade winds south of 20°N (Zenk et al., 1991),

and becomes a narrow strip of less than 100 km width in winter (Mittlelstaedt, 1983). The MC

advances onto the shelf during summer and is enhanced by the relatively strong Equatorial

Countercurrent and the southerly monsoon (Mittelstaedt, 1983). The presence of strong coastal

currents during the upwelling season causes substantial horizontal shear within the surface layer,

where currents tend to converge (Mittelstaedt, 1983). This convergence favors the formation of the

140 Cape Verde Frontal Zone (CVFZ, Fig. 1; Zenk et al., 1991).

A coastal countercurrent, the Poleward Undercurrent (PUC; Fig. 1) occurs mainly due to wind-

142 driven offshore divergence (Pelegrí et al., 2017). North of Cape Blanc (ca. 21°N), the intense





northeasterly winds cause the coastal upwelling to move further offshore and the upper slope is

144 filled with upwelling waters. South of Cape Blanc (Fig. 1), northerly winds dominate year through

but surface waters remain stratified and the PUC occurs as a subsurface current. South of Cape

Timiris (ca. 19°30'N), the PUC intensifies during summer-fall and remains at the subsurface during

winter–spring (Pelegrí et al., 2017). The encountering of the northward flowing MC-PUC system

with the southward flowing currents in the Canary Basin leads to flow confluence at the CVFZ

(Zenk et al., 1991) and the offshore water export visible as the giant Mauritanian chlorophyll

filament (Gabric, 1993; Pelegrí et al., 2006; Pelegrí et al., 2017). This filament extends over 300

km offshore  (*e.g*., Van Camp et al., 1991; Arístegui et al., 2009; Cropper et al., 2014) and carries

South Atlantic Central Water (SACW) offshore through an intense jet-like flow (Meunier et al.,

2012; Fig. 1). Intense offshore transport forms an important mechanism for the export of cool,

nutrient-rich shelf and upper slope waters offshore Mauritania. Based on satellite imagery and *in

situ* data, it has been estimated that the giant Mauritanian filament could export about 50% of the

156 particulate coastal new production offshore toward the open ocean during intervals of most intense

upwelling, while coastal phytoplankton at the surface might be transported as far as 400 km

offshore (Gabric et al., 1993; Barton, 1998; Lange et al., 1998; Helmke et al., 2005). The transport

effect could extend to even more distant regions in the deep ocean, since sinking particles are

160 strongly advected by lateral transport (Fischer and Karakaș, 2009; Karakaș et al., 2006, Fischer et

al., 2009).

The nutrient concentration of the upwelled waters off Mauritania varies depending on their origin

(Fütterer, 1983; Mittelstaedt, 1991; Zenk et al., 1991). The source of upwelling waters off

Mauritania are either North Atlantic Central Water (NACW), north of about 23°N, or SACW (south

of 21°N, Fig. 1). Both water masses are mixed in the filament area off Cape Blanc. The SACW

occurs in layers between 100 and 400 m depth off Cape Blanc and the Banc d'Arguin. The

hydrographic properties of the upwelling waters on the shelf suggest that they ascend from depths

between 100 and 200 m south off the Banc d'Arguin (Mittelstaedt, 1983). North of it, the SACW

merges gradually into deeper layers (200-400 m) below the CC (Mittelstaedt, 1983). During intense

upwelling, the stratification of the shelf waters weakens, and so is the stratification further offshore,

usually within the upper 100 m (Mittelstaedt, 1991). The biological response is drastically



accelerated in the upwelling waters when the SACW of the upper part of the undercurrent feeds

the onshore transport of intermediate layers to form mixed-water types on the shelf (Zenk et al.,

1991).

**3. Material and Methods**

**3.1. Moorings, sediment traps and fluxes**

Sediment trap moorings were deployed at site CBeu off Mauritania in the CC-EBUE (Fig. 1;

Table 1). Details on trap depth, sample number, and sampling intervals as well as the sample

identification (cruise and GeoB numbers) are presented in Table 1. Large-aperture time-series

sediment traps of the Kiel type with 20 to 40 cups (depending on ship-time availability, Table 1)

and 0.5 m² openings, equipped with a honeycomb baffle (Kremling et al., 1996), were used. Traps

were moored in intermediate waters (between 1,256 m and 1,296 m) and sampling intervals varied

between 6.5 and 23 days (Table 1). Uncertainties with the trapping efficiency due to strong

currents (*e.g.* undersampling, Buesseler et al., 2007) and/or due to the migration and activity of

zooplankton migrators ('swimmer problem') are assumed to be minimal in this depth range.

     Prior to each deployment, sampling cups were poisoned with 1 ml of concentrated $HgCL_2$ per

100 ml of filtered seawater. Pure NaCl was used to increase the density in the sampling cups up to

40 ‰. Upon recovery, samples were stored at 4°C and wet-split in the MARUM sediment trap

laboratory (University Bremen, Bremen) using a rotating McLANE wet splitter system. Larger

swimmers, such as crustaceans, were handpicked with forceps and removed by carefully filtering

through a 1-mm sieve. All flux data hereafter refer to the size fraction of <1 mm. Detailed

information about sampling and laboratory analysis is given in Mollenhauer et al. (2015) where the

bulk fluxes are given for the deployments CBeu 1-4. Additionally to the fluxes, alkenone derived

sea surface temperature (SST) for the CBeu deployments 1-4 were provided by these authors.

Using ¼ or $^{1}/_{5}$ wet splits, analysis of the <1 mm fraction was carried out (Fischer and Wefer,

1991; Fischer et al., 2016). Samples were freeze-dried and homogenized before being analyzed

for bulk (total mass), organic carbon (OC), calcium carbonate ($CaCO_3$) and biogenic silica (BSi,

opal). Total organic carbon (TOC) and $CaCO_3$ were measured by combustion with a CHN-Analyzer

(HERAEUS, Dept. of Geosciences, University of Bremen). TOC was measured after removal of





carbonate with 2 N HCl. Overall analytical precision based on internal lab standards was better

than 0.1% (±1σ). Carbonate was determined by subtracting OC from total carbon, the latter being

measured by combustion without pre-treatment with 2N HCl. Organic matter was estimated by

multiplying the content of total organic carbon by a factor of two as about 50-60% of marine

organic matter is constituted by OC (Hedges et al., 2002).

BSi was determined with a sequential leaching technique with 1M NaOH at 85°C (Müller and

Schneider, 1993). The precision of the overall method based on replicate analyses is mostly

between ±0.2 and ±0.4%, depending on the material analyzed. The lithogenic fluxes were

estimated by subtracting the masses of $CaCO_3$, BSi, and 2 x OC from the total mass flux.

**3.2. Assessment of organism fluxes and species identification**

**3.2.1. Diatoms**

For this study 1/25 and 1/125 splits of the original samples were used. Samples were prepared

for diatom studies following the method proposed by Schrader and Gersonde (1978). A total of 185

sediment trap samples were processed. Each split was treated with potassium permanganate,

hydrogen peroxide, and concentrated hydrochloric acid following previously used methodology

(Romero et al., 2002, 2009a, b, 2016, 2017). Identification and count of the species assemblage

were done on permanent slides (*Mountex*® mounting medium) at x1000 magnification using a

*Zeiss*®Axioscop with phase-contrast illumination (MARUM, Bremen).  The counting procedure and

definition of counting units follows Schrader and Gersonde (1978). Depending on valve

abundances in each sample, several traverses across each slide were examined. Total amount of

counted valves per slide ranged between 300 and 800. At least two cover slips per sample were

scanned in this way. Valve counts of replicate slides indicate that the analytical error of

concentration estimates is ca. 10 %. The resulting counts yielded abundance of individual diatom

taxa (absolute and relative) as well as daily fluxes of valves per m-[2] d-[1], calculated according to

Sancetta & Calvert (1988).

**3.2.2. Coccolithophores**

Aliquots of each sample were 1/125 of the <1 mm fraction. Depending on the total flux, samples

were further split down to 1/625 to 1/2500 of the original sample volume and were filtered

afterward onto polycarbonate membrane filters (Schleicher and Schuell[TM] 47mm diameter, 0.45µm



pore size). A piece of the filter was cut and mounted on a Scanning Electron Microscopy (SEM)

stub. Qualitative and quantitative analyses of the trapped assemblages were performed using a

232 *Zeiss*® DSM 940A at 10kV accelerating voltage (Department of Geosciences, University of

Bremen, Bremen). In an arbitrarily chosen transect, coccoliths were counted until a total of at least

234 500 specimens were reached. The taxonomic classification of identified species was based on

Young et al. (2003) as well as on Nannotax 3 (Young et al., 2019).

**3.2.3. Organic-walled and calcareous dinoflagellate cysts**

1/125 splits of the original trap material was ultrasonically treated and sieved with tap water

through a high precision metal sieve (Storck-Veco®) with a 20µm pore size. The residue was

transferred to Eppendorff cups and concentrated to 1 ml of suspension. After homogenization of

240 the material, a known aliquot was transferred to a microscope slide where it was embedded in

glycerin-gelatine, covered with a cover slip and sealed with wax to prevent oxidation of the

242 organic material. After counting, cyst fluxes were calculated by multiplying the cyst counts with

the aliquot fraction and the split size (1/125) and dividing through the amount of days during

which the trap material was sampled as well as the trap-capture surface. No chemicals were

used to prevent dissolution of calcite and silicate. Cyst assemblages were determined by light

microscopy (Axiovert, x400 magnification). Taxonomy of organic walled dinoflagellate cysts is

according to Zonneveld and Pospelova (2015), taxonomy of calcareous dinoflagellate cysts is

248 according to Vink et al. (2002) and Elbrächter et al. (2008)

**3.2.4. Planktonic foraminifera and pteropods**

Depending on the absolute magnitude of the total mass flux, a 1/5 or a 1/25 split of the wet

solution (fraction <1mm) was used to pick planktonic foraminifers and pteropods (pelagic

mollusks). Specimens of both groups of calcareous microorganisms were rinsed three times by

using tap water, dried at 50°C in an oven overnight and then separated from each other.

Identification and count of shells were done by using a stereomicroscope *Zeiss*® Stemi 2000

(Marum, Bremen). The foraminifera fluxes (all size fractions) were determined in mg per m$^2$ and

256 day with a Sartorius BP 211D analytical balance. Additionally, the total amount of

specimens/sample of foraminifera and pteropods (>150 µm) were counted manually. Foraminifera

were identified and classified according to Hemleben et al. (1989) and Schiebel and Hemleben



(2017). Out of 15 species of planktonic foraminifers identified, only six species were used as
environmental indicators.

**3.2.5. Alkenones**

1/5 wet splits of the <1mm fraction were used for alkenone analysis. Briefly, freeze-dried CBeu

1-4 samples were solvent extracted. The resulting total lipid extracts (TLEs) saponifiedn  and the

alkenone fractions were obtained by means of column chromatography of the neutral lipid fractions

from the saponification. Details are given in Mollenhauer et al. (2015).

A slightly different, miniaturized analysis procedure has been applied for the CBeu trap 5

samples. 1/5 wet splits of the freeze-dried <1mm fraction were weighted in 10 ml Pyrex tubes and

a known amount of an internal standard (*n*-Nonadecan-2-one) was added. Samples were then 3x

ultrasonically extracted with a mixture of 3 ml dichloromethane/methanol (9:1 vol./vol.), centrifuged

and the supernatant solvent combined as total lipid extract (TLE). TLEs were evaporated to

dryness and saponified in a 0.1M potassium hydroxide solution in methanol/water (9:1 vol./vol.) for

two hours at 80°C. Neutral lipids, recovered with hexane, were afterwards separated into fractions

of different polarity by silica gel chromatography and elution with hexane, dichloromethane/hexane

(1:1 vol./vol.) and dichloromethane/methanol (9:1 vol./vol.), respectively. The second fraction

containing the alkenones was dried, re-dissolved in 20µl hexane and analyzed on a 7890A gas

chromatograph (GC, Agilent Technologies) equipped with a cold on-column injection system, a

DB-5MS fused silica capillary column (60 m, ID 250 µm, 0.25 µm film coupled to a 5 m, ID 530 µm

deactivated fused silica precolumn) and a flame ionization detector (FID). Helium was used as

carrier gas (constant flow, 1.5 mL/min) and the GC oven was heated using the following

temperature program: 60 °C for 1 min, 20 °C/min to 150 °C, 6 °C/min to 320 °C and a final hold

time of 35 min. Alkenones were identified by comparison of the retention times with a reference

sample composed of known compounds. Peak areas were determined by integrating the

respective peaks.

The $U_{37}^{K'}$ index was calculated using the following equation (Prahl and Wakeham, 1987)

$$U_{37}^{K'} = \frac{C_{37:2}}{C_{37:2}+C_{37:3}}$$





and converted to SST using the global surface water calibration from Conte et al. (2006):

$$SST = \frac{U_{37}^{K'} - 0.0709}{0.0322}$$

**3.3. Environmental physical parameters**

SST, Sea Surface Temperature Anomaly (SSTA), mixed layer depth (MLD) and upper ocean

chlorophyll-a concentration data are based on satellite-derived data achieved from the NASA

supported Giovanni project (https://giovanni.gsfc.nasa.gov/). SST is the mean of daily surface

ocean temperature and MLD values of the sampling interval in a 4km$^2$ area around the trap

position (Table 1). In the research area, SST at the trap position is influenced by seasonal air

temperature changes as well as the presence of upheld water surfacing at the trap position.

To compensate for seasonal air temperature changes the SSTA is calculated by subtracting the

above-calculated SST at the trap position from mean SST values of simultaneous sampling

intervals in a 4km$^2$ block 200nm west of the trap position. Both SSTA and MLD are parameters

reflecting active upwelling in the study area. Upper ocean chlorophyll *a* data and MLD represent

monthly mean values in a 9km$^2$ block around the trap position. Wind speed and wind directions are

provided by Nouadhibou airport (20°56′N, 17°2′W) (Institut Mauretanien de Recherches

Océanographiques et des Pêches, Nouadhibou, Mauritania). For statistical analyses, the means of

daily values during the trap sampling intervals were calculated.

**3.4. Multivariate analyses**

The ordination techniques Principal Component (PCA) and Redundancy (RDA) analyses have

been performed with the software Package Canoco 5 (ter Braak and Smilauer, 2012; Smilauer and

Leps, 2014). To obtain insights into the temporal relationship between fluxes of organism groups

(diatoms, coccolithophores, organic-walled dinoflagellate cysts, calcareous dinoflagellate cysts,

planktonic foraminifera and pteropods) and bulk components as well as the environmental

conditions in surface waters and low atmosphere a RDA has been performed. RDA compares the

total flux of organism groups with environmental parameters and TOC, BSi, CaCO$_3$ and lithogenic

fluxes (Table 2). Since the fluxes of the individual groups differ by several orders of magnitudes, it

is essential to normalize their flux values prior to the statistical analysis in order to be able to

determine temporal relationships of flux variability. As a consequence, the total flux of the





organism groups have been normalized to values between 0 and 1000 previous to the analyses

according to formula 1:

$nFl_i = (FL_{i/y} / FL_i \, max) \times 1000n$

$Fl_i$ = normalized flux of species group i

           $Fl_{i/y}$ = flux of species group i in sample y

$FL_i \, max$ = maximal flux observed in species group i

        To better understand the relationship within the individual organism groups, a PCA has been

performed (Table 2). For these analyses, the total flux of the organisms/species groups have been

normalized to values between 0 and 1000 according to formula 2:

$nFl_j = (FL_{ij/y} / FL_i \, max) \times 1000n$

           $Fl_j$ = normalized accumulation rate of ecological entity j in species group i

$Flj_{/y}$ = accumulation rate of ecological entity j in sample $yFL_j \, max$ = maximal accumulation

           rate observed in species group i

Within coccolithophores, *Umbilicosphaera anulus* had exceptionally large fluxes in one sample

only. This flux exceeded the maximal flux of the other species by a factor of three. This value has

been excluded from the analysis and the $FL_j \, max$ in this group is determined by excluding this

outlier.

**4. Results**

**4.1. Bulk fluxes and fluxes of organism groups**

        On average, the carbonate fraction ($CaCO_3$) dominates the mass flux (41% to the total mass

flux) and is mainly composed of coccolithophores, foraminifera, calcareous dinoflagellates and

pteropods (see also Fischer et al., 2009, 2016). $CaCO_3$ is followed by BSi (average = 14.5%,

mostly diatoms, Romero and Fischer, 2017), and organic carbon (6.5%, delivered by diatoms,

coccolithophores and organic dinoflagellate cysts). The lithogenic fraction –mostly composed of

mineral dust– makes up 31.5% of the total mass for the entire sampling period of CBeu 1-5 (2003-

2008, Table 1). Bulk fluxes for the CBeu deployments 1 to 4 were already presented in

Mollenhauer et al. (2015; Table 1) in combination with fluxes of the lipid fraction and SST



reconstructions. The SST record is extended with new alkenone data until March 2008 (CBeu

deployment 5, Table 1).

The fluxes of total mass, CaCO$_3$, TOC, BSi and lithogenics show major peaks in winter and

spring (Fig. 2). Secondary maxima were found during late summer/fall, mainly in 2003, and less

clear in 2005, 2006 and 2007 (Fig. 2). However, the individual components reveal different flux

amplitudes and point to some interannual variability. Carbonate fluxes were exceptionally high in

early winter 2005 compared to the other years. Fluxes of BSi and organic carbon match well the

total flux pattern and show less interannual variability (Fig. 2c, d). The flux of the lithogenic fraction

has the highest amplitudes in spring 2006 and 2007 (Fig. 2e).

Fluxes of microorganisms are dominated by diatoms and coccoliths (Fig. 3a, b). These

exceeded the fluxes of organic- and calcareous walled dinoflagellate cysts, planktonic foraminifera

and pteropods by a factor of four to five. Highest coccoliths and diatom fluxes reach 4.2 x 10$^9$

coccoliths m$^{-2}$ d$^{-1}$, and 1.2 x 10$^8$ valves m$^{-2}$ d$^{-1}$, respectively. Maximal fluxes of organic-walled

dinoflagellates reach up to 7.1 x 10$^4$ cysts m$^{-2}$ d$^{-1}$, and of planktonic foraminifera 0.9 x 10$^4$ shells

m$^{-2}$ d$^{-1}$, and 1.1 x 10$^4$ pteropods shells m$^{-2}$ d$^{-1}$.

Each group of organisms shows large seasonal and interannual variabilities. Diatoms had their

maximal flux in fall/winter 2005 and spring/summer 2006 (Fig. 3a). Coccolithophores had their

highest export fluxes mostly in winter/spring throughout the sampling interval and exceptionally in

July/August 2003 and 2007 and in fall 2005. On the long-term, low coccolithophore fluxes are

observed fall and winter 2007/2008 (Fig. 3b). Calcareous dinoflagellate cysts were practically

absent until fall 2005 (Fig. 3c). After September 2005, calcareous dinoflagellate cysts showed

maximal export fluxes in fall/winter 2005/2006 and fall/winter 2006/2007 (Fig. 3c). Fluxes

decreased again after spring 2007. Organic-walled dinoflagellate cysts had their highest export

fluxes in summer 2003, spring/summer 2006 and summer 2007 (Fig. 3d). Planktonic foraminifera

showed maximal fluxes in summer 2003, winter/spring 2004, 2005, 2007 and spring/summer 2006

(Fig. 3e). Pteropods had their maximal fluxes in summer 2003, fall/winter 2003/2004, 2004/2005

and 2006/2007 as well as summer 2005 and 2007 (Fig. 3f).

**4.2. Species- and group-specific composition of assemblages**





The studied plankton community at the CBeu site is highly diverse and is composed by at least

220 identified species. Table 3 presents the species-specific composition of groups depicted in Fig.

4.

Out of 170 marine **diatom** species, the 70 most abundant diatom taxa (average relative

contribution >0.75% for the entire studied interval) were attributed in four groups, according to the

main ecological conditions they represent: (1) benthic, (2) coastal upwelling, (3) coastal planktonic,

and (4) open-ocean waters (see also Romero and Fischer, 2017). The diatom groups show a clear

seasonal pattern (Fig. 4a) with benthic diatoms having higher relative contributions during spring

and summer, whereas the coastal upwelling group mainly occurred between late spring and early

fall. Open-ocean diatoms were more abundant from fall to early spring while the coastal planktonic

taxa tended to be more abundant during fall and winter. Most noticeable, a drastic shift in the

relative contribution of the benthic diatoms occurred in spring-summer 2006 when the abundance

of benthic diatoms strongly increased from 2006 onward, compared to 2003–2005 (Fig. 4a). In

spite of the increased relative contribution of benthic diatoms after 2005, the seasonal pattern of

the predominantly high spring-summer total diatom flux remained unaltered (Fig. 3a).

**Coccolithophores** are consistently dominated by *Emiliania huxleyi* and *Gephyrocapsa*

*oceanica*, whose contribution is always higher 50% of the community throughout the sampling

period (Fig. 4b). Oligotrophic upper photic zone (UPZ, *e.g.*, *Umbellosphaera tenuis*, *U. irregularis*)

and lower photic zone species (LPZ, *e.g.*, *Florisphaera profunda*, *Gladiolithus flabellatus*) make up

the majority of the remaining species. Whereas *E. huxleyi* showed a less clear seasonal pattern,

*G. oceanica* tends to be more abundant during late spring and early fall (Fig. 4b). In contrast, UPZ

and LPZ taxa have higher relative contributions during winter and spring. The appearance of

*Umbilicosphaera anulus* (present in consistently low relative abundances of 5-10% until the

summer of 2006) accounts for up to 65% of the community in winter 2005/06. Other common taxa

with an average relative contribution >0.75% for the entire studied interval are listed in Table 3.

**Calcareous dinoflagellates** can be attributed to five groups based according to the main

ecological conditions they represent (Siggelkow et al., 2002; Richter et al., 2007; Kohn and

Zonneveld, 2010); (1) upwelling, (2) warm waters, (3) terrestrial mineral input, (4) cosmopolitan

and (5) other species (Table 3). Until fall 2005 abundances are very low such that the



Plankton variability off Mauritania

recognition of a seasonal pattern is hampered (Fig. 4c). After fall 2005, their occurrence shows a

more distinguished seasonal pattern. In spring-summer of 2006 upwelling species dominate the

association. After fall 2006, the community is composed by the interplay of cosmopolitan

species, warm water taxa and upwelling-dependent species, where warm water taxa dominate.

Whereas upwelling species are most abundant in spring and fall, warm water and mineral

indicators are more abundant in fall/winter (Fig. 4).

***Organic dinoflagellates*** can be attributed to five groups based on the relationship between

their geographic distribution in surface sediments from the Cape Blanc area and the

environmental conditions in surface and subsurface waters as well as long-term surveys of their

seasonal cyst production (Susek et al., 2005; Holzwarth et al., 2010; Smayda, 2010; Smayda

and Trainer, 2010; Trainer et al., 2010; Zonneveld et al., 2012, 2013): (1) upwelling, (2)

upwelling relaxation, (3) potential toxic, (4) cosmopolitan, and (5) other species.  Throughout the

investigated time interval upwelling species are abundant in spring and fall/winter whereas

upwelling relaxation species have higher relative abundances in fall (Fig. 4d). Potential toxic

species are abundant in fall/winter 2004/2005 and 2007/2008. Organic-walled dinoflagellate

cysts do not show a clear change in their composition between 2005 and early 2006 as

observed for many other groups.

The distribution and abundance of ***planktonic foraminifera*** species is linked to surface-water

properties. We use prominent species as tracers of surface water properties: *Globigerina bulloides*

(upwelling species) is generally most abundant between summer and fall (fig. 4e). *Globorotalia*

*inflata* and *Neogloboquadrina incompta* (transitional and subpolar species) are present mostly

throughout, only decreasing in abundance in fall and winter when warm water taxa peaked

(*Globigerinoides ruber* pink, *G. ruber* white and *G. sacculifer*; Kucera, 2007, Schiebel and

Hemleben, 2017) (Fig. 4e). The only exception is in fall and winter 2004/2005, when warm-water

taxa are almost absent.

As large secondary carbonate producers off Mauritania, ***pteropods*** are important contributors

to the carbonate flux in the CC-EBUE (Fischer et al., 2016). The community is composed of

relatively few taxa. *Heliconoides inflatus* (formerly known as *Limacina inflata*) dominates the

assemblage throughout most of the studied interval (Fig. 4f). It is often the only species found in





the assemblage until winter 2005/2006, when a sudden and drastic shift in the relative contribution

occurred. *Limacina bulimoides* appears for the first time in winter to spring 2006 - and again in fall

and winter 2006/2007 - and dominates the assemblage together with a group of unspecified

uncoiled pteropods. However, another occurrence of *L. bulimoides* is missing in winter 2007/2008.

**4.3. Statistical analyses**

Comparison of the fluxes of the microorganism groups, bulk fluxes and the environmental

conditions in surface waters and the lower atmosphere (MLD, average wind speed, wind direction,

chlorophyll-*a* concentration (Chl-*a*), SST and SSTA) resulted in a significant relationship within the

first and second RDA axes that correspond to 34 % and 11% of the variance within the dataset,

respectively (Table 2).

All microorganism groups are ordinated at the positive part of the first axis showing a positive

relationship with all bulk parameters (Fig. 5). This implies that the fluxes of all studied

microorganisms groups increase with increasing fluxes of total mass, TOC, lithogenic, BSi and

$CaCO_3$ (Fig. 5). Fluxes of planktonic foraminifera, diatoms and –to a lesser extent–

coccolithophores and organic dinoflagellates are ordinated at the negative site of SST and, with

exception of organic dinoflagellates, positive side of MLD (Fig. 5). This implies that their fluxes are

enhanced whenever SST is low and MLD is deep, i.e. under a well-mixed uppermost water

column. Diatoms, coccolithophores, organic dinoflagellates and planktonic foraminifera also show

a positive correlation with SSTA, implying that enhanced fluxes of these microorganisms occur

when temperature anomalies between waters overlying site CBeu and the offshore pelagial is

large. The fluxes of pteropods and calcareous dinoflagellate cysts are positively related to the

average wind direction, and negatively to MLD and average wind speed (Fig. 5).

To better understand the correlation of the fluxes of the species groups within the organism

groups, PCA has been performed (Fig. 6, Table 2). The first two axes correspond to 26.3 % and

16.2% of the variance within the dataset respectively. Based on their ordination on the first and

second axis, three groups are recognized (Fig. 6):

-    Groups 1 and 2 are ordinated at the negative side of the second axis. Group 1 (in blue, Fig.

6) is built by planktonic foraminifera characteristic of cooler or upwelled water masses (For-

cold, For-upw); benthic and upwelling-related diatoms (Dia-bent and Dia-upw); organic





dinoflagellates characteristic for upwelling regions (OD-upw), and the coccolithophore

*Gephyrocapsa oceanica* (Co-*Gocean*). Group 2 (in brown, Fig. 6) consists of upwelling–

related and other calcareous dinoflagellates cysts (CD-upw and CD-other), other

coccolithophores (Co-other), and coastal planktonic and open-ocean diatoms (Dia-coast

and Dia-ocean).

-    Ordinated at the positive side of the second axis and central part of the first axis, group 3

assembles planktonic foraminifera mainly thriving in warm waters (For-warm), calcareous

dinoflagellates characteristic of warm water conditions and those responding to mineral

input (CD-warm, CD-min), and all pteropods groups or species (in black, Fig. 6).

-    Group 4 is ordinated at the central/positive part of the second axis and positive site of the

first axis. Species assigned to group 4 are: organic walled dinoflagellate cysts typical of the

upwelling relaxation (OD-upw relax); UPZ and LPZ coccolithophores (Co-up phot and Co-

low phot); *E. huxleyi* (Co-*Ehux*), other coccolithophores (Co-other), *U. anulus* (Co-*Uanu*),

and cosmopolitan calcareous dinoflagellate cysts (CD-cosm) (in red, Fig. 6).

**5. Discussion**

**5.1. Relationship between microorganisms fluxes at site CBeu and the physical and**

**biogeochemical settings off Mauritania**

Both the visual examination of the flux variability and the statistical analysis document that the

seasonality of most microorganism groups at the CBeu site closely follows the temporal pattern of

changes in upper water oceanographic conditions off Mauritania between June 2003 and March

2008. Fluxes of diatoms, coccolithophores, organic-walled dinoflagellate cysts and planktonic

foraminifera increase whenever the uppermost water column is well mixed (Fig. 7e), SSTs are low

(Fig. 7d), and SSTA are high (Figs. 2, 3, 5). This strong match supports the scenario of

simultaneous occurrence of intense upwelling off Mauritania and high microorganisms fluxes and

production at site CBeu. Several previous studies have separately documented enhanced

production of diatoms, coccolithophores, organic-walled dinoflagellate cysts and planktonic

foraminifera occurring when the nutrient concentration in the uppermost water column off

Mauritania increases (Baumann et al., 2005; Zonneveld et al., 2012; Guerreiro et al., 2017;



Romero and Fischer, 2017; Pospelova et al., 2018; Jiménez-Quiroz et al., 2019).

The atmospheric, hydrographic and biochemical conditions deliver the physical and nutrient

frames that determine the temporal pattern of population dynamics as recorded by the CBeu trap.

Wind and upper water conditions off Mauritania show a clear seasonal pattern of variability (Fig.

7a-e). The highly stratified uppermost water column (above 40 m water depth) overlying site CBeu

is an effect of winds blowing mainly from the N-NE between late winter and early summer (Fig. 7a,

b, e). The stratification breaks down mostly in early to middle winter, when the predominant winds

turn from N-NE into S-SE (Fig. 7a). Following this setting, upwelling off Mauritania reaches its

highest intensity between late winter/early spring and early summer (Mittelstaedt, 1991; Meunier et

al., 2012; Cropper et al., 2014). The SST record (Fig. 7d) matches well the seasonal evolution of

stratification and mixing conditions: lowest temperatures mostly between winter and early spring

(increasing SSTA in late winter and throughout the spring season). Throughout the period

investigated, this SST cyclicity remains fairly constant.

Fluxes of total mass and biogenic bulk components (TOC, BSi, $CaCO_3$) are clearly seasonal in

nature (Fig. 3 a-d; Fischer et al., 2019) and reflect the temporal productivity pattern of the

Mauritanian upwelling region (Meunier et al., 2012; Cropper et al., 2014). The good temporal

match between maxima of most of the studied microorganism groups and biogenic bulk

components unambiguously evidences the contribution of primary and secondary producers to the

total mass/biogenic mass fluxes off Mauritania (Figs. 2, 3). Higher absolute values of $CaCO_3$ over

BSi (Fig. 2b, d) support the scenario of calcareous primary and secondary producers

(coccolithophores, foraminifera and pteropods) dominating the plankton community in the

Mauritanian upwelling system (Fischer et al., 2019). Diatoms are the main contributors to the BSi

flux (Fig. 3a, 2d; Romero et al., 2002; Romero and Fischer, 2017).

A strong match among fluxes of diatoms, coccolithophores, organic-walled dinoflagellate cysts

and planktonic foraminifera with lithogenic fluxes at times of enhanced upwelling is observed (Figs.

2e, 3 a-c, e). The RDA supports this correlation (Fig. 5). The good correlation between lithogenic

and microorganisms fluxes demonstrates that winds –responsible for the water column mixing off

Mauritania (Mittelstaedt, 1983; Meunier et al., 2012)– might additionally enrich surface waters

overlying site CBeu with land-derived nutrients. Primary and secondary producers may remarkably



benefit from this eolian-transported pool of nutrients. Lithogenic material is brought into

Mauritanian ocean waters in the form of dust that it is transported from the Sahara and the Sahel

(Romero et al., 2003; Friese et al., 2017). Numerous studies have thoroughly documented that the

particle flux off Mauritania predominantly occurs in the form of aggregates, often rich in lithogenic

particles (*e.g.*, Karakaş et al., 2009; Iversen et al., 2010; Iversen and Ploug, 2010; Nowald et al.,

2015; Fischer et al., 2016; van der Jagt et al., 2018). Recent experiments have also shown that

aggregates' abundance and sinking velocities increase toward deeper waters when aggregates

are ballasted with lithogenic particles, whereas aggregates are not able to scavenge lithogenic

material from deeper waters (van der Jagt et al., 2018).

A remarkable finding of our multiannual trap experiment is that flux maxima of diatoms,

coccolithophores, organic-walled dinoflagellate (all primary producers) and planktonic foraminifera

(secondary producers) seem to occur fairly simultaneously (Figs. 3, 5). We propose three possible

interpretations: (*i*) no clear short-term succession of the microorganism groups occurred (no

temporal turnover in phytoplankton composition within a few days, Roelke and Spatharis, 2015),

(*ii*) the succession is not properly captured due to low temporal resolution of some sediment trap

intervals (Table 1), and/or (*iii*) the microorganisms –originally produced in surface and subsurface

waters by different communities– sink with different velocities through the water column toward the

ocean bottom and get 'mixed' during their sinking, mainly due to dissimilar weights and sizes of

their remains. However, the high-resolution intervals of CBeu deployments 4 and 5 (up to 7.5 days

per sample, Oct 2006-March 2008, Table 1) should have captured a possible short-term

succession of major groups (*e.g.*, diatoms quickly reacting to increasing nutrient availability,

whereas photosynthetic dinoflagellates becoming more abundant during upwelling relaxation,

Margalef, 1963; Jiménez-Quiroz et al., 2019). Although we do not observe a clear pattern of

succession within studied populations, at this stage we do not disregard either its occurrence. It

should be kept in mind that the deployed traps capture those microorganism remains that reach

the trap cups at around 1,300 m water depth, while they do not capture green algae or

cyanobacteria thriving in surface waters. CBeu traps at ca. 1,300 m water depth capture a mixed

signal of sinking particles from a surface catchment area of at least ca. 100 km$^2$ (Siegel and

Deuser, 1997, Fischer et al., 2016,) due to (*i*) differential settling velocities of particles (Fischer and





Plankton variability off Mauritania

Karakaş, 2009; Iversen et al., 2010, van der Jagt et al., 2018), and (*ii*) highly heterogeneous and

dynamic surface water conditions due to filament and eddy activity off Mauritania (Mittelstaedt, 1991; Gabric et al., 1993; Meunier et al., 2012; Cropper et al., 2014). Additionally, the trapped

signal is always affected by dissolution of particular species and/or groups of organisms sinking through the water column intro deeper waters (*e.g*., Romero et al., 1999).

**5.2. Temporal variations of the species-specific composition of the plankton community**

We are aware that 1,900 days of continuous sampling cannot deliver a definite picture of all

temporal changes affecting the composition of the plankton community in the very dynamic Mauritanian upwelling. However, the overall temporal pattern observed let us to propose a general

sequence of seasonal variability. Most of the major microorganisms groups occur simultaneously and clear successional trends are not quite distinguishable (Fig. 3). A consistent seasonal pattern

in the occurrence of species or groups of species is yet recognized. Figure 4 shows the seasonal evolution of populations responding to the temporal dynamics of nutrient availability, *e.g*. following

short-period dust events (Fig. 2e) and/or vertical mixing events associated with stronger winds (Fig. 7a, e). Based on the visual data examination and the statistical analysis, four groups of

species are recognized (Figs. 3, 6). Populations of group 1 (Dia-bent, Dia-upw, Co-*Gocean*, OD-upw, For-cold = in blue in Fig. 6) have higher relative contribution during the most intense phase of

the upwelling season (mainly between late winter/early spring and early summer; Mittelstaedt, 1983, Cropper et al., 2014). Group 1 quickly responds to intense mixing and lowered SST at the

CBeu site (Fig. 7d, e) and represents the typical upwelling-related association off Mauritania. This observation confirms the ecological characterization of the species groups that has been

separately presented in previous biogeographical/ecological studies (Romero et al., 2002; Kucera, 2007; Köbrich et al., 2008, 2016; Zonneveld et al., 2013; Romero and Fischer, 2017).

Diatoms of coastal regions (Dia-coast, non-upwelling related) and those thriving in open ocean waters (Dia-ocean) together with other calcareous dinoflagellates (CD-other), cosmopolitan

organic dinoflagellates (OD-cosm) and 'other coccolithophores' (Co-other) are assigned to group 2 (in brown in Fig. 6). Except for the cosmopolitan organic dinoflagellates cysts, all components of

group 2 are primary producers and occur more abundantly between early fall and late winter (Fig. 4), at times of deepening of the ML and upwelling relaxation (Fig. 7e). Group 2 represents a





primary producers signal typical of meso- to oligotrophic waters conditions off Mauritania, occurring

under weakened upwelling, when winds predominantly blow from the N-NE, SST start decreasing

after their summer peak, and the uppermost water column stratifies (Fig. 7a, d, e).

Except for warm waters (CD-warm) and dust input-sensitive (CD-min) calcareous

dinoflagellates cysts, group 3 is mainly composed by secondary producers: warm-water planktonic

foraminifera and all pteropods (Fig. 4e, f). As such, this group represents the calcareous fraction of

zooplankton feeding on other (primary) phytoplankton, occurring mainly during phases of

predominantly warmer SSTs (Fig. 7d), N-NE-originated winds (Fig. 7a) and stratified uppermost

water column (Fig. 7e). The distribution and abundance of planktonic foraminifera species is

strongly linked to surface-water properties. SST appears to be the most important factor controlling

assemblage composition of planktonic foraminifera (Kucera, 2007). Large, symbiont-bearing

specialists like *Globigerinoides ruber and G. sacculifer* are adapted to more oligotrophic and

warmer waters. They show their maximum abundance in warm waters with a deeper mixed-layer

depth (Fig. 7e,f).

The seasonal dynamics of group 4 is similar to that of group 3 (intervals of weakened upwelling

conditions), but they differ in their composition: group 4 is mainly made of calcareous primary

producers. These populations dominate the plankton community during intervals of weakened

upwelling, shallow MLD and predominantly oligotrophic water conditions. Similar to group 3, group

4 consists mainly of coccolithophores (the dominant *E. huxleyi*, accompanied by UPZ and LPZ, *U.

anulus*, Figs. 4b, 6), as well as organic dinoflagellate cysts characteristic for upwelling relaxation

phases (CD-upw relax). The contribution of *E. huxleyi* and accompanying coccolithophore taxa,

and upwelling-relaxation organic dinoflagellate cysts shows highest relative values from early fall

through early spring and decreases into the most intense upwelling season (when *G. oceanica*

increases, Fig. 4b). As such, this group also bears some resemblance to group 2, though coastal

and open-ocean water diatoms are component of the latter, while diatoms are absent in group 4.

This difference possibly reflects the distinct nutrient and water depth conditions in which *E. huxleyi*

and other coccolithophores (group 4) and diatoms (group 2) typically thrive.

**5.3. Shifts in the species-specific composition of assemblages between 2004 and 2006**





The persistent seasonal pattern of the groups' and species occurrence experiences occasional

shifts. Several events, which altered the 'regular' pattern of temporal occurrence of species or

group species at site CBeu, were observed between late 2004 and late 2006 (Fig. 7f-j). We identify

three main shift stages in the species-specific composition of assemblages:

1.    Stage 1 (2004): (*i*) low total biogenic production (summer–fall 2004, Fig. 2b-d), and (*ii*)

absence of warm-water foraminifera (Fig. 7f). These changes in production/flux were

accompanied by (*iii*) a significant decrease in SST as reconstructed with $U^{K\prime}_{37}$ (Fig. 7d).

2.    Stage 2 (late 2005/early 2006): (*i*) extraordinarily high relative contribution of the

coccolithophore *U. anulus* (Fig. 7i); as well as (*ii*) the first high occurrence of *L. bulimoides*

and uncoiled pteropods (Fig. 7h).

3.    Stage 3 (after fall 2006): (*i*) strong increase of the relative contribution of benthic diatoms

(Fig. 7i) and warm-water calcareous dinoflagellates (Fig. 7j), and (*ii*) highest longest

occurrence and highest relative abundance of *L. bulimoides* (fall 2006/winter 2007, Fig.

7h).

A certain degree of interannual variability of the physical setting (Mittelstaedt, 1983, 1991;

Cropper et al., 2014) might explain the shifts in the species-specific composition of the

assemblages. The almost disappearance of warm-water planktonic foraminifera in 2004 (Fig. 7f)

was most probable the response to lower-than-usual water temperatures (Fig. 7d). However, the

SST decrease is not recorded by satellite imagery. The overall climate evolution indicates a longer

warm and dry period from 2001-2004 in the Sahel and Sahara (east of site CBeu) and

anomalously warm temperatures in the Eastern Atlantic (Zeeberg et al., 2008; Alheit et al., 2014).

2004 is the only year of our study with the largest lag between satellite and $U^{K\prime}_{37}$-based temperature

(Fig. 7d). This temperature gap suggests a certain decoupling between the temperature signal of

the uppermost centimeters of the water column (satellite) and subsurface waters where the

alkenone-forming coccolithophores dwell (*E. huxleyi* and *G. oceanica*; Conte et al., 1995). As

planktonic foraminifera mainly react to SST variability (Kucera, 2007), cooler than usual subsurface

waters between middle winter and early fall 2004 (Fig. 7d) might have been responsible for the

strong decrease of the warm-water planktonic foraminifera contribution (Fig. 7f). Additionally, all

other plankton groups show lowest fluxes toward late summer. Neither the seasonal pattern nor





the MLD show any significant change nor unusual high fluxes of lithogenic occurred (Fig. 2e, 7e).

Exceptionally, the winter season 2004/2005 is characterized by a high total flux (Fig. 2a); this

extraordinarily high seasonal value matches well highest fluxes of TOC and $CaCO_3$ for the studied

interval.

The extraordinary high relative abundance of *U. anulus* in fall 2005 has not yet been observed

in similar or other settings, although it is often listed in studies of large-scale distribution patterns of

coccolithophores (*e.g.*, Böckel and Baumann, 2008; Estrada et al., 2016; Poulton et al., 2017). So

far only Steinmetz (1991) has found *U. anulus* (described as *U. calvata* and *U. scituloma*) in

'frequent' abundances in sediment traps deployed in the equatorial Atlantic, central Pacific, and in

the Panama Basin, but without adding appropriate information such as fluxes, the timing of its

occurrence or its ecological significance. In most of earlier trap studies, *U. anulus* has been

grouped together with other umbilicosphaerids coccolithophores, since it did not reached high

abundances (*e.g.*, Köbrich et al., 2016; Guerreiro et al., 2017). Nevertheless, umbilicosphaerids

seem to favor warm and more oligotrophic conditions (Baumann et al., 2016), so that the increased

input of tropical surface waters transported northward via the MC (Mittelstaedt, 1991) can be

possibly responsible for the advection of *U. anulus* upon the CBeu site.

The shift in the pteropod composition from dominating *H. inflatus* towards the appearance of *L.*

*bulimoides* between winter 2005 and spring 2006 –and again in fall and winter 2006/2007 (Fig.

7h)– can be also explained by the increased influence of warmer surface waters of southern origin.

*Heliconoides inflatus* is known as a rather cosmopolitan species, occurring across a wide range of

oceanic provinces (Bé and Gilmer, 1977; Burridge et al., 2017), whereas *L. bulimoides* seems to

prefer waters of subtropical gyres (although it was also present in low numbers in the equatorial

region, Burridge et al., 2017). A stronger transport of the MC from the south may have led to the

deterioration of the adequate environmental conditions for *H. inflatus*, as can be seen by the

extremely low total pteropods flux during winter 2005 to spring 2006 (Fig. 3e), and, thus, to the

relative enrichment of *L. bulimoides*. The fact that the latter species is absent again in winter 2008

(Fig. 7h) represents a gradual return to previous ('regular') winter conditions. 'Regular'

temperatures from early 2005 on allowed the reappearance of warm-water planktonic foraminifera

in fall 2005 (Fig. 7f).


658 The outstanding increase in the contribution of the benthic diatoms in spring-summer 2006 (Fig. 7i) might have been possibly related to the intensification of lateral advection upon the

660 intermediate-waters deployed CBeu trap (Romero and Fischer, 2017). Observational and model experiments show that the transport of particles from the Mauritanian shelf and the uppermost

662 slope via nepheloid layers significantly contributes to the deposition upon the lowermost slope and beyond than the direct vertical settling of particles from the surface layer (Nowald et al., 2014;

664 Karakaş et al., 2006; Fischer et al., 2009; Zonneveld et al., 2018). The relevance of advective processes within nepheloid layers has been already proposed for similar settings (Puig and

666 Palanques, 1998; Inthorn et al., 2006). We speculate that the longer predominance of N-NE winds between 2005 and 2007 (Fig. 7a) might have possibly intensified the transport of benthic diatoms

668 from the shallow coastal area into the hemipelagic CBeu trap via the MC (Fig. 1). Enhanced lateral transport has important environmental implications for the final burial of organic matter in EBUEs.

670 As the organic matter can be effectively displaced from the area of production (Inthorn et al., 2006), carbon depocenters generally occur at the continental slopes between 500 and 2,000 m. In

672 the CC-EBUE around Cape Blanc, the depocenter with up to 3% of organic carbon has a depth range between 1,000 and 2,000m (Fischer et al., 2019).

674 Most of the populations affected by and responding to shifting environmental conditions off Mauritania between 2004 and 2006 returned to their 'regular' seasonal pattern of occurrence after

676 2006 (Fig. 4). However, some shifts persisted still after summer 2006. *Limacina bulimoides* still dominated the pteropod assemblage (Fig. 7h), the total pteropod flux showed the highest maxima

678 for the entire studied interval (might be due to the large food supply and organic matter as represented by high total fluxes of diatoms, Fig. 3a, e), and warm-water calcareous dinoflagellate

680 cyst increased during late fall 2006 (Fig. 7j). An exception to this pattern is the high relative contribution of benthic diatoms (Figs. 4a, 7i; Romero and Fischer, 2017). At this stage, we cannot

682 fully disregarded that the shift in the species-specific composition of the diatom community (also present after 2008; Romero and Fischer, 2017; Romero, unpublished observations) might be due

684 to the natural long-term variability due to external forcings (*e.g.*, North Atlantic Oscillation) or due to on-going climate change.

686 Our multiannual trap experiment provides a unique opportunity to study the long-term evolution





of the plankton community in an ecologically important EBUE. Rapid shifts in the population

contribution at the trap site CBeu demonstrate that calcareous, siliceous and organic plankton

microorganisms rapidly react to environmental changes in the CC-EBUE off Mauritania. Time-

series trap experiments continuously conducted over many years –as those currently in the CC-

EBUE (Fischer et al., 2016, 2019; Romero et al., 2002, 2016, 2017)– deliver a reliable

observational basis on the occurrence of long-lasting variations of populations in response to key

environmental forcings. Among others, our multiannual observations will be useful for future model

experiments on plankton dynamics and evolution in low- and mid latitude EBUEs and how

organisms influencing the global carbon cycle might react to global and ocean warming.

**6. Conclusions**

-   The seasonal amplitude of the flux variations of primary and secondary producers in the upper

water column off Mauritania is well recorded in our 1,900 days continuous trap experiment. The

repeated yearly pattern of higher fluxes of diatoms, coccolithophores, organic-walled dinoflagellate

cysts and planktonic foraminifera between early spring and early/middle summer match well the

temporal occurrence of the most intense upwelling interval in waters overlying the trap site CBeu.

Instead, fluxes of calcareous dinoflagellate cysts and pteropods are higher during intervals of

upwelling relaxation (late summer through late fall).

-   The good temporal match between maxima of (most of) studied microorganism groups and

biogenic bulk components unambiguously evidences the contribution of primary and secondary

producers to the total mass/biogenic mass fluxes. The notorious coupling between fluxes of

lithogenics and major microorganisms groups provides compelling evidence for the biological

pump off Mauritania to be strongly dependent on the dust input from the Sahara/Sahel and the

eolian-transported nutrient deposition.

-   1,900 days of continuous trap record of microorganism fluxes let recognizing a general

sequence of seasonal variations of the main plankton populations thriving in coastal waters off

Mauritania. The temporal turnover (succession) is better shown by the temporal variations of

particular species or group of species.



- A significant shift in the 'regular' seasonal pattern of populations' occurrence is recognized in species relative contributions between 2004 and 2006. Several events altering the regular seasonal pattern were observed and occurred in three main stages: summer–fall 2004, late

2005/early 2006, and after fall 2006. Although most of the populations return 'to normal' after fall 2006, a few did not.

- Our multiannual trap experiment emphasizes the significance of long-term records on evaluating the impact of changing environmental conditions on living populations. Time-series trap

experiments conducted over many years –as those currently conducted in the CC-EBUE by MARUM– deliver a broad observational basis on the occurrence of persistent seasonal pattern as

well long-lasting variations of microorganisms changes in response to key forcings, such as nutrient input, water masses variability, lateral transport and/or climate change. The applicability of

the flux dynamics of primary and secondary producers here presented is not limited to the Mauritanian upwelling system, and it might comparable to other EBUEs.

**Code and Data Availability**

Data are available at https://doi.pangaea.de/10.1594/PANGAEA.904390

**Author Contributions**

     All authors collected the data. Oscar E. Romero wrote the manuscript. All authors contributed to

results interpretation and discussion.

**Competing Interests**

     The authors declare that they have no conflict of interest.

**Acknowledgements**

We are greatly indebted to the masters and crews of the RVs Poseidon and MS Merian. We greatly appreciate the help of the RV Poseidon headquarters at Geomar (K. Lackschewitz, Kiel,

Germany) during the planning phases of the research expeditions (Table 1) and the support by the German, Moroccan and Mauritanian authorities in Berlin, Rabat and Nouakchott. We also thank





the IMROP and its director at Nouadhibou (Mauritania) for their general support and the help in

getting the necessary permissions to perform our multiyear trap experiments in Mauritanian coastal

waters. We thank G. Ruhland, N. Nowald and M. Klann (MARUM, Bremen) for mooring

deployments and lab work on the samples (Table 1). This work was possible due to the long-term

funding by the German Research Foundation (DFG) through SFB 261, the Research Center

Ocean Margins (RCOM) and the MARUM Excellence Cluster "The Ocean in the Earth System"

(University of Bremen, Bremen, Germany).

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





Plankton variability off Mauritania

**Figures**

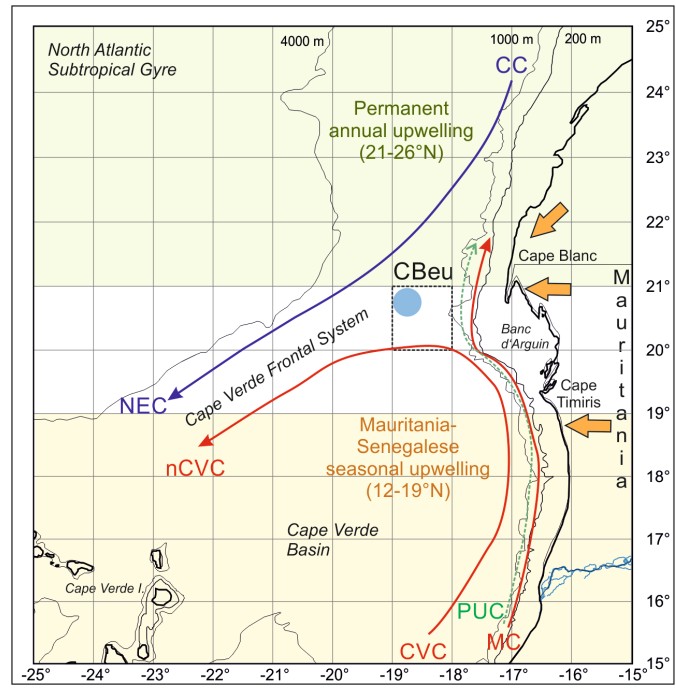

Figure 1. Map of the study area showing the location of trap site CBeu (full light blue dot), surface
currents, and main wind system. Surface currents (Canary Current, CC, violet line; North
       Equatorial Counter Current, blue arrow; Mauritanian Current; red arrow), North Equatorial
Current (NEC), Cape Verde Current (CVC), north Cape Verde Current (nCVC), PUC are
       depicted after Mittelstaedt (1983, 1991) and Zenk et al. (1991). The Cape Verde Frontal Zone
(CVFZ) builds at the confluence of the NACW and the SACW (Zenk et al., 1991). Trade winds
       and Saharan Air Layer are represented by orange arrows (Nicholson, 2013). The upwelling
zones are depicted after Cropper et al. (2014).



Plankton variability off Mauritania

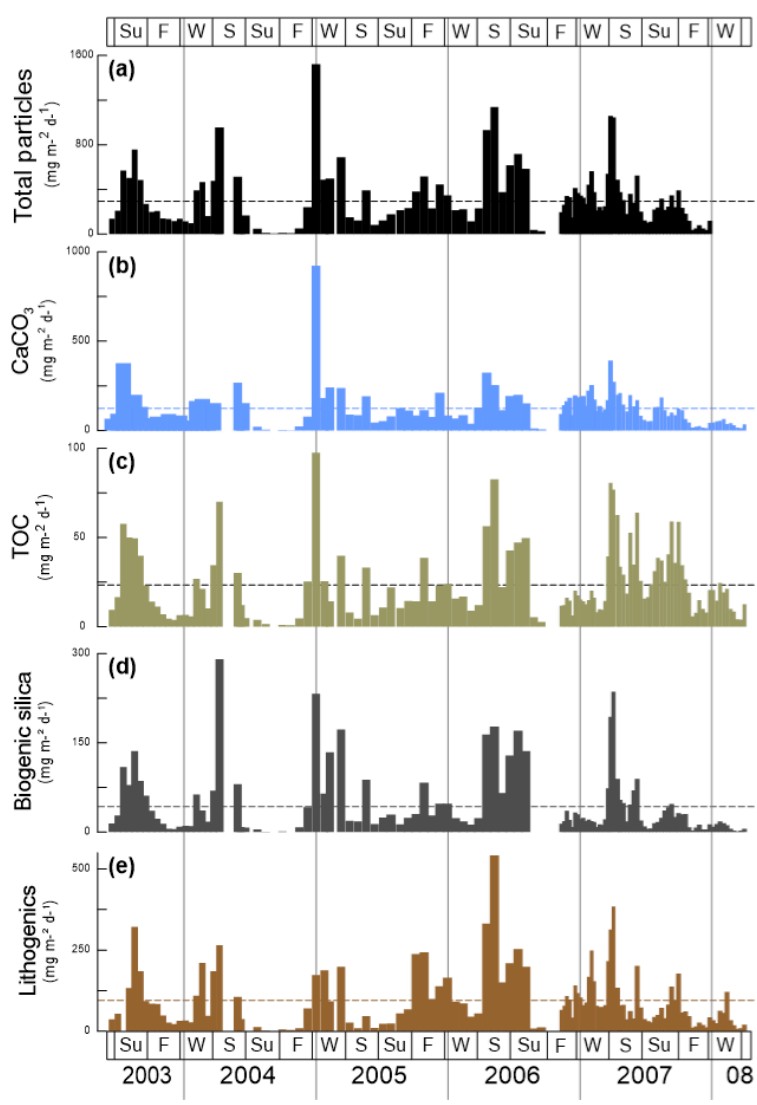

Figure 2.  Total particle and bulk fluxes at the trap site CBeu between June 2003 and March 2008.
From top to bottom: (a) total particle (mg m$^{-2}$ d$^{-1}$, black bars), (b) calcium carbonate (CaCO$_3$,
mg m$^{-2}$ d$^{-1}$, light blue bars), (c) total organic carbon (TOC, mg m$^{-2}$ d$^{-1}$, olive bars), (d) biogenic
silica (BSi, opal, mg m$^{-2}$ d$^{-1}$, dark grey bars), and (e) lithogenics (mg m$^{-2}$ d$^{-1}$, brown bars). The
horizontal stripped line for each parameter represents the average flux for the whole studied
interval (see Table 1). The boxes in the upper and lower panels represent seasons
(Su=summer, F=fall, W=Winter, S=spring). The vertical background gray lines indicate calendar
year separation. For interpretation of the references to color in this figure legend, the reader is
referred to the web version of this article.



Plankton variability off Mauritania

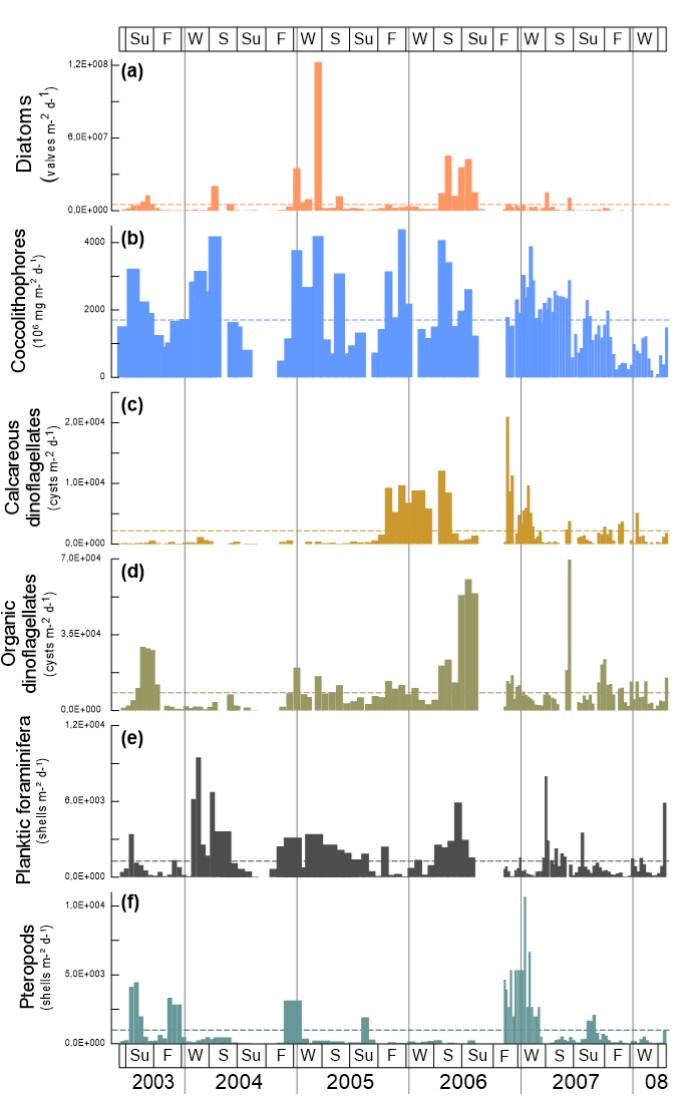

Figure 3. Fluxes of microorganisms at the trap site CBeu between June 2003 and March 2008.
From top to bottom: (a) diatoms (valves m$^{-2}$ d$^{-1}$, peach bars; note that ten samples corresponding to CBeu 5 –12/13/2007 through 03/17/2008– were not available for diatom
analysis); (b) coccolithophores (coccoliths m$^{-2}$ d$^{-1}$, light blue bars); (c) calcareous dinoflagellates (cysts m$^{-2}$ d$^{-1}$; gold bars); (d) organic dinoflagellates (cysts m$^{-2}$ d$^{-1}$; khaki bars); (f) planktic
foraminifera (shells m$^{-2}$ d$^{-1}$; grey bars), and (f) pteropods (shells m$^{-2}$ d$^{-1}$; ocean green bars). The horizontal stripped line for each group of organisms represents the average flux for the whole
study interval. The boxes in the upper and lower panels represent seasons (Su=summer, F=fall, W=Winter, S=spring). The vertical background gray lines indicate calendar year separation. For
interpretation of the references to color in this figure legend, the reader is referred to the web version of this article.





Plankton variability off Mauritania

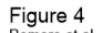

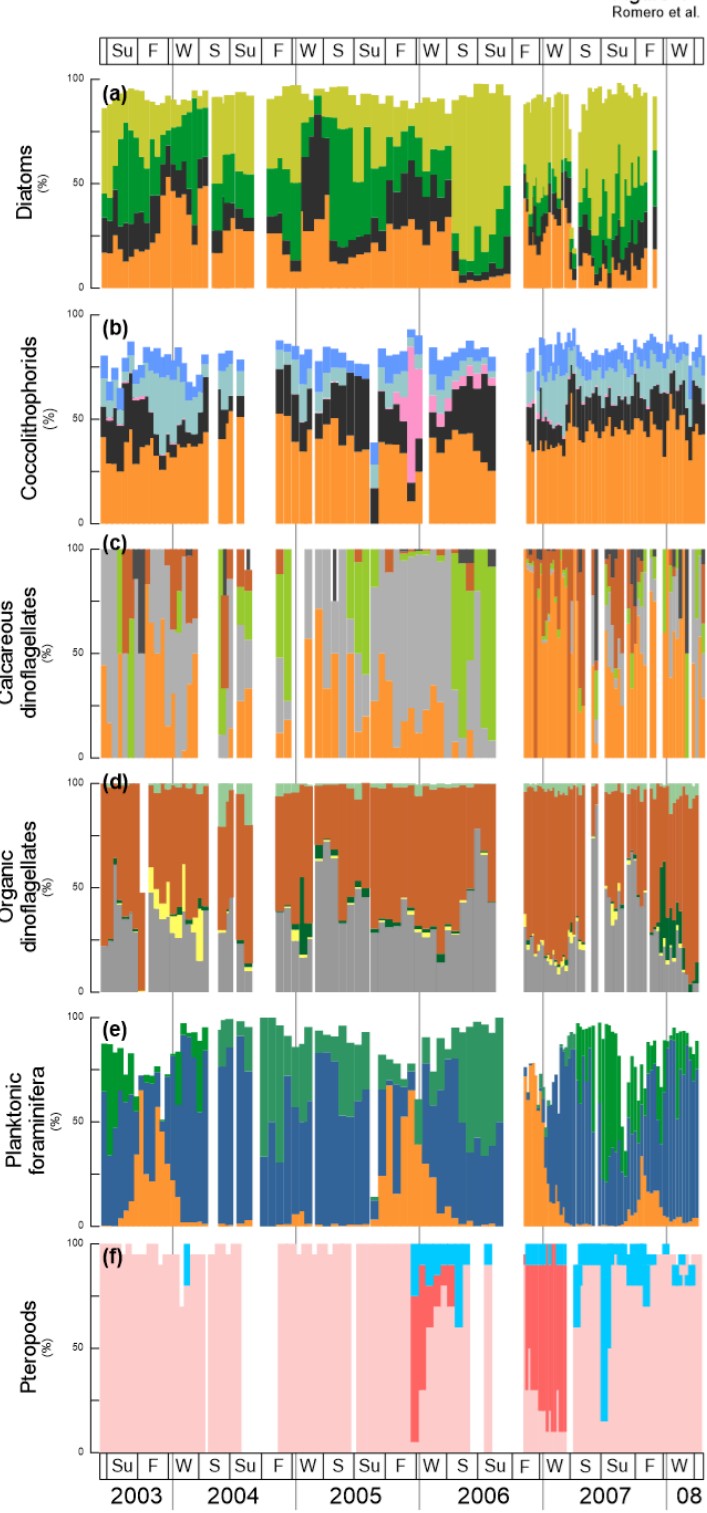



Figure 4. Cumulative relative abundance (%) of main species or group of species of
diatoms, coccolithophores, dinoflagellates, planktonic foraminifera and pteropods at the trap
site CBeu between June 2003 and March 2008 (Table 1). From top to bottom: (a) diatoms -
benthic, light green bars; coastal upwelling, dark green bars; coastal planktonic, black bars;
and open-ocean (%, orange bars; note that ten samples corresponding to CBeu 5 –12/13/2007
through 03/17/2008– were not available for diatom analysis); (b) coccolithophores –upper
photic zone, blue bars; lower photic zone, moss green; *Umbilicosphaera anulus*, pink bars;
*Gephyrocapsa oceanica*, black bars; *Emiliana huxleyi*, orange bars; (c) calcareous
dinoflagellates – other calcareous, dark grey bars; mineral-input related, brown bars; upwelling,
light green bars; cosmopolitan, light grey; warm water, orange bars; (d) organic dinoflagellates
– upwelling species (grey bars); upwelling relaxation species (light yellow bars); potential toxic
(dark green bars); cosmopolitan, red brown bars; other, faded green bars; (e) planktic
foraminifera – upwelling, green bars; cool water, blue bars; warm water, orange bars; and (f)
pteropods – uncoiled species, light blue bars; *Limacina bulimoides*, red bars; *Heliconoides
inflatus*, pink bars. The species-specific composition of groups is presented in Table 3. The
boxes in the upper and lower panels represent seasons (Su=summer, F=fall, W=Winter,
S=spring). The vertical background gray lines indicate calendar year separation. For
interpretation of the references to color in this figure legend, the reader is referred to the web
version of this article.


Figure 5
Romero et al.

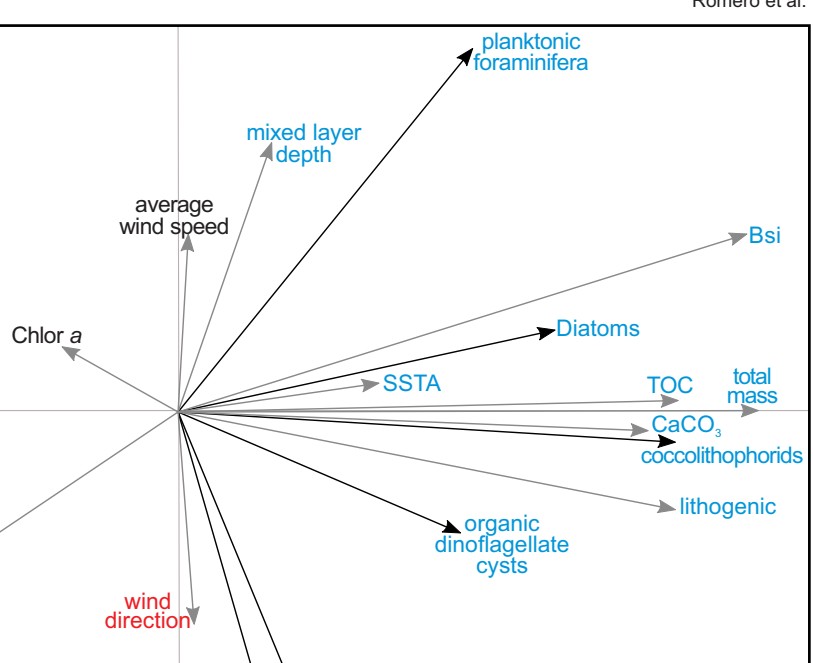

Figure 5. RDA ordination diagram depicting the relationship between the accumulation rates of organism groups and bulk fluxes and environmental conditions in upper waters. References: Av. wind speed=average wind speed; Chlor-*a*=chlorophyll *a*; TOC=total organic carbon; CaCO$_3$=calcium carbonate; mixed layer depth; SST=sea surface temperature; SSTA=sea surface temperature anomalies. For interpretation of the references to color in this figure legend, the reader is referred to the web version of this article.


Figure 6
Romero et al.

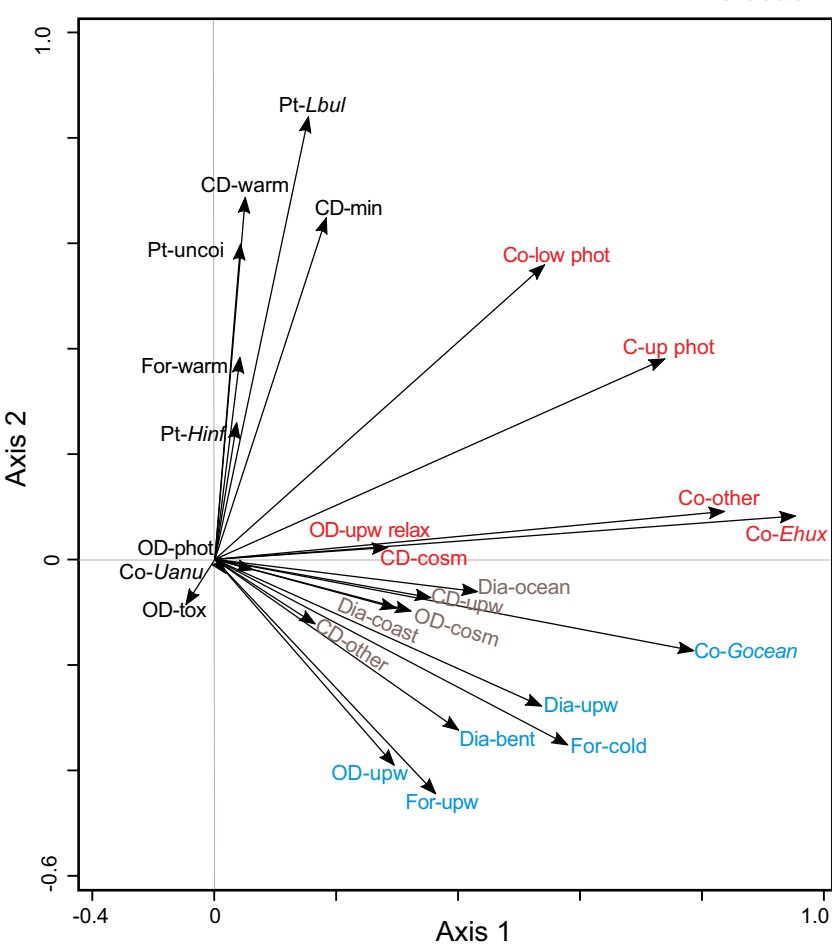

Figure 6. Results of a PCA analysis of ecological groups of the organism groups at the CBeu trap site between June 2003 and March 2008. References: Dia (diatoms): -bent = benthic, -coast = coastal planktonic, -ocean = open ocean, -upw = upwelling; Co {coccolithophores): *-Ehux* = *Emiliana huxleyi*, *-Gocean* = *Gephyrocapsa oceanica*, -low phot: low photic zone, -other: other coccoclithophroids, *-Uanu* = *Umbilicosphaera anulus,* -up phot: upper photic zone; CD (calcareous dinoflagellate cysts): -cosm = cosmopolitan group, -min = terrestrial mineral group, -other = species that do not fit in one of the other ecological groups, -upw = upwelling, warm: warm waters; OD (organic-walled dinoflagellate cysts): -cosm = cosmopolitan group, -other = species that do not fit in one of the other ecological groups, -tox = potential toxic group, -upw = upwelling, -upw relax = upwelling relaxation; For (foraminifera): -cold: cold water group, -upw = upwelling group; -warm = warm water group; and Pt (pteropods): *-Hinf* = *Heliconoides inflatus*, *-Lbul* = *Limacina bulimoides*, -uncoi: uncoiled. Groups of microorganisms are identified by colors (light blue, group 1; brown, group 2; black, group 3; and red, group 4). The species-specific composition of groups is presented in Table 3. For interpretation of the references to color in this figure legend, the reader is referred to the web version of this article.



Plankton variability off Mauritania

Figure 7
Romero et al.





Figure 7. Comparison physical data (a-e) and relative abundance of selected species or group/s
      of species (f-j) at site CBeu between June 2003 and March 2008. Physical data: (a) daily wind
direction (°, the grey line are daily data, the thicker black line represent the 17-running points
      average); (b) daily wind velocity (m s$^{-1}$, the grey line are daily data, the thicker dark brown line
represent the 17-running points average), (c) dust storm event (number of events) recorded at
      the meteorological station of the Airport of Nouadhibou (ca. 20°57'N, 17°02'W, Mauritania); (d)
seawater temperature (°C): $U^{K'}_{37}$-based reconstruction (black line) and satellite-imagery
      generated data (blue line); (e) mixed layer depth (m, grey line, https://modis.gsfc.nasa.gov for
the area between 19°-18°W and 20°-21°N). Relative contribution (%) of (f) planktonic warm-
      water foraminifera (orange bars); (g) the coccolithophore *A. anulus* (pink bars); (h) pteropods *L.
bulimoides* (red bars) and uncoiled species (light blue bars); (i) benthic diatoms (banana yellow
      bars); and (j) calcareous dinoflagellates (light orange bars). The species-specific composition of
groups is presented in Table 3. The vertical gray lines indicate years separation. The light grey
      shading in the background highlights the interval of main shift in fluxes values and/or the
relative contribution of particular species or group of species (see discussion in 5.3.). For
      interpretation of the references to color in this figure legend, the reader is referred to the web
version of this article.





**Tables**

Table 1: Data deployment at site CBeu (Cape Blanc eutrophic, off Mauritania): coordinates, GeoB
location and cruise, trap depth, sample amount, capture duration of each sample and sampling
interval. Two gaps in the sampling intervals occurred: 04/05/2004–04/18/2004, and 09/28/2006–
10/28/2006.

| Mooring CBeu | Coordinates | GeoB-#/ cruise | Trap depth (m) | Sample amount | Capture duration (sample/days) | Sampling interval |
|---|---|---|---|---|---|---|
| 1 | 20°45'N 18°42'W | - POS 310 | 1,296 | 20 | 1 = 10.5, 2-20 = 15.5 | 06/05/2003 – 04/05/2004 |
| 2 | 20°45'N 18°42'W | 9630-2 M 65-2 | 1,296 | 20 | 1-20 = 22, 2-19 = 23 | 04/18/2004 – 07/20/2005 |
| 3 | 20°45.5'N 18°41.9'W | 11404-3 POS 344- | 1,277 | 20 | 21.5 | 07/25/05 – 09/28/2006 |
| 4 | 20°45.7'N 18°42.4'W | 11835-2 MSM 04b | 1,256 | 20 | 1 = 3.5, 2-20 = 7.5 | 10/28/2006 – 03/23/2007 |
| 5 | 20°44.9'N 18°42.7'W | 12910-2 POS 365- | 1,263 | 38 | 1, 2 = 6.5, 3-38 = 9.5 | 03/28/2007 – 03/17/2008 |

Table 2: Main result values of the ordination techniques Redundancy (RDA) and Principal
Component (PCA) analyses performed with the software Package Canoco 5 (ter Braak and
1226 Smilauer, 2012; Smilauer and Leps, 2014).

| Analysis | Method | Analysed Parameters | Length of gradient | Eigenvalue Axis 1 (%) | Eigenvalue Axis 2 (%) | Eigenvalue Axis 3 (%) | Eigenvalue Axis 4 (%) | P-value |
|---|---|---|---|---|---|---|---|---|
| 1 | RDA | Fluxes of microorganisms and bulk parameters, environmental parameters | 1.8 | 34.5 | 10.7 | 4.7 | 2.1 | 0.002 |
| 2 | PCA | microorganisms | 1.4 | 26.3 | 16.2 | 9.8 | 6.9 | |

References: RDA, Redundancy Analysis; PDA, Principal Component
Analysis.





Table 3: Species composition of the groups of (a) diatoms, (b) coccolithophores, (c) calcareous
       and (d) organic dinoflagellate cysts, (e) planktonic foraminifera and (f) pteropods at Site CBeu
between June 2003 and March 2008.

| Diatoms | References |
|---|---|
| **1) Benthic** | |
| *Actinoptychus* spp. | Round et al. (1990) |
| *Amphora* spp. | |
| *Cocconeis* spp. | |
| *Cymatosira belgica* | |
| *Delphineis surirella* | |
| *Grammatophora marina* | |
| *Licmophora* sp. | |
| *Odontella mobiliensis* | |
| *Psammodyction panduriformis* | |
| *Tabullaria* spp. | |
| **2) Coastal upwelling** | |
| Resting spores of: | Hasle and Syvertsen (1996) |
| *Chaetoceros affinis* | |
| *Chaetoceros cinctus* | |
| *Chaetoceros compresus* | |
| *Chaetoceros constrictus* | |
| *Chaetoceros coronatus* | |
| *Chaetoceros debilis* | |
| *Chaetoceros diadema* | |
| *Chaetoceros radicans* | |
| **3) Coastal planktonic** | |
| *Actinocyclus curvatulus* | Crosta et al. (2012), Romero et |
| *Actinocyclus octonarius* | al. (2009, 2012, 2016, 2017), |
| *Actinocyclus subtilis* | Romero and Armand (2010) |
| *Chaetoceros concavicornis* (vegetative cell, VC) | |
| *Chaetoceros lorenzianus* (VC) | |
| *Chaetoceros pseudobrevis* (VC) | |
| *Coscinosdiscus argus* | |
| *Coscinosdiscus decrescens* | |
| *Coscinosdiscus radiatus* | |
| *Cyclotella litoralis* | |
| *Skeletonema costatum* | |
| *Thalassionema nitzschioides* var. *capitulata* | |
| *Thalassiosira angulata* | |
| *Thalassiosira conferta* | |
| *Thalassiosira oestrupii* var. *venrickae* | |
| *Thalassiosira poro-irregulata* | |
| **4) Open-ocean** | |
| *Asteromphalus flabellatus* | Hasle and Syvertsen (1996), |
| *Asteromphalus sarcophagus* | Romero et al. (2005), Crosta et |
| *Azpetia neocrenulata* | al. (2012) |
| *Azpetia nodulifera* | |
| *Azpetia tabularis* | |
| *Detonula pumila* | |
| *Dytilum brightwellii* | |
| *Fragilariopsis doliolus* | |
| *Hemiaulus hauckii* | |
| *Hemidiscus membranaceus* | |
| *Leptocyclindrus mediterraneus* | |
| *Neodelphineis denticula* | |
| *Nitzschia bicapitata* | |
| *Nitzschia capuluspalae* | |
| *Nitzschia interruptestriata* | |
| *Nitzschia sicula* | |





*Planktoniella sol*
*Pseudo-nitzschia inflata* var. *capitata*
*Pseudo-nitzschia pungens*
*Pseudo-nitzschia subfraudulenta*
*Pseudosolenia calcar-avis*
*Pseudotriceratium punctatum*
*Rhizosolenia acuminata*
*Rhizosolenia bergonii*
*Rhizosolenia imbricatae*
*Rhizosolenia setigera*
*Roperia tessellata*
*Stellarima stellaris*
*Thalassionema bacillare*
*Thalassionema frauenfeldii*
*Thalassionema nitzschioides* var. *capitulata*
*Thalassionema nitzschioides* var. *inflata*
*Thalassionema nitzschioides* var. *parva*
*Thalassiosira eccentrica*
*Thalassionema endoseriata*
*Thalassiosira ferelineata*
*Thalassiosira lineata*
*Thalassiosira nanolineata*
*Thalassiosira oestrupii var. oestrupii*
*Thalassiosira sacketii var. sacketii*
*Thalassiosira sacketii var. plana*
*Thalassiosira subtilis*
*Thalassiosira symmetrica*

**Coccolithophores**

| | |
|---|---|
| 1) Cosmopolitan | Boeckel and Baumann (2008), Baumann and Boeckel (2013), Poulton et al. (2017), Young et al. (2019) |
| *Emiliania huxleyi* | |
| *Gephyrocapsa oceanica* | |

2) Lower photic zone
*Algirosphaera robusta*
*Calciosolenia murrayi*
*Florisphaera profunda*
*Gladiolithus flabellatus*
*Hayaster perplexus*
3) Warm oligotrophic surface waters
*Discosphaera tubifera*
*Helicosphaera carteri*
*Rhabdosphaera xiphos*
*Umbellosphaera irregularis*
*Umbellosphaera tenuis*
*Umbilicosphaera anulus*
*Umbilicosphaera sibogae*
4) Other miscellanous species
*Acanthoica quattrospina*
*Calcidiscus leptoporus*
*Calcidiscus leptoporus* small
*Calcidiscus quadriperforatus*
*Gephyrocapsa ericsonii*
*Gephyrocapsa muellerae*
*Ophiaster hydroideus*
*Ophiaster hydroideus*
*Rhabdosphaera stylifer*
*Syracosphaera anthos*
*Syracosphaera pulchra*

**Calcareous dinoflagellates cysts**

| | |
|---|---|
| 1) Upwelling | Siggelkow et al. (2002); Richter et al. (2007); Kohn and |
| *Calciodinellum operosum* | |

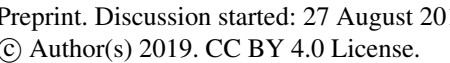 

|  |  |
|---|---|
| *Scrippsiella trochoidea* | Zonneveld (2010) |

2) Warm waters
    *Calciodinellum albatrosianum*

3) Mineral input
    *Leonella granifera*

4) Cosmopolitan
    *Thoracosphaera heimii*

5) Others
    *Calciodinellum levantinum*
    *Melodemuncula berlinensis*
    *Pernambugia tuberosa*
    *Scrippsiella lacrymosa*
    *Scrippsiella regalis*
    *Scrippsiella trifida*

**Organic dinoflagellates cysts**

| | |
|---|---|
| 1) Upwelling<br>    *Echinidinium aculeatum*<br>    *Echinidinium granulatum*<br>    *Echinidinium transparantum*<br>    *Echinidinium zonneveldii*<br>    *Ecginidinium* spp.<br>    cyst of *Protoperidinium americanum*<br>    cyst of *Protoperidnium monospinum*<br>    *Stelladinium stellatum* | Susek et al. (2005); Holzwarth et al. (2010); Trainer et al. (2010); Smayda (2010); Smayda and Trainer (2010); Zonneveld et al. (2010; 2013) |

2) Upwelling relaxation
    *Lingulodinium machaerophorum*
    cyst of *Polykrikos schwarzii*
    cyst of *Polykrikos kofoidii*

3) Potential toxic
    cysts of *Gymnodinium* spp.

4) Cosmopolitan
    *Brigantedinium* spp.
    *Spiniferites* species
    *Impagidinium* species

**Planktonic Foraminifera**

| | |
|---|---|
| 1) Warmer waters<br>    *Globigerinoides ruber* (pink and white)<br>    *Globigerinoides sacculifer* | Hemleben et al. (1989), Schiebel and Hemleben (2017) |

2) Cooler waters
    *Globorotalia inflata*
    *Neogloboquadrina incompta*

3) Upwelling
    *Globigerina bulloides*

4) Additional secondary species
    *Beella digitata* (Brady 1879)
    *Globigerinella calida* (Parker 1962)
    *Globigerinella siphonifera* (d'Orbigny 1839)
    *Globorotalia crassaformis* (Galloway and Wissler 1927)
    *Globorotalia menardii* (Parker, Jones and Brady 1865)
    *Globorotalia scitula* (Brady 1882)
    *Globorotalia truncatulinoides* (d'Orbigny 1839)
    *Neogloboquadrina dutertrei* (d'orbigny 1839)
    *Orbulina universa* (d'Orbigny 1839)
    *Pulleniatina obliquiloculata* (Parker and Jones 1865)

**Pteropods**

| | |
|---|---|
| *Heliconoides inflatus* (d'Orbigny 1835)<br>*Limacina bulimoides* (d'Orbigny 1835) | WoRMS Editorial Board (2017) |