# Peer review of "Flux variability of phyto- and zooplankton communities in the Mauritanian coastal upwelling between 2003 and 2008"

_Biogeosciences, 2019_

## Referee Comment (RC1) · Manuel Bringué (Referee) · 4 Oct 2019

The manuscript by Romero et al. presents a unique dataset of a wide array of planktonic organisms, as recorded nearly continuously over a period of $\sim$ 5 years, at a site off Cape Blanc. Such multi-annual, high-resolution records are extremely rare, and the authors take advantage of data on multiple major planktonic groups generated over several years, to integrate the findings and inform on pelagic food web dynamics, the ecology of functional groups, and how environmental conditions affect planktonic organisms – and in turn, how those 'surface' signals can be encoded into the flux of sinking particles. I suspect a lot of effort was applied to the classification of each

species into environmentally significant groups, which, to my knowledge, seems accurate. Overall, this study represents a great contribution to the current state of knowledge on planktonic food webs, and findings can be applied in other similar settings (e.g., upwelling systems) around the world.

The manuscript is a great fit for BG; the text and figures are of very good quality, analyses were well designed and I agree with most of the author's interpretations. My evaluation is overwhelmingly positive and I only have a few minor comments that I would like to see discussed/addressed before final publication of the manuscript, as detailed below.

General comments One aspect I would like to see discussed further is the possibility of the trap record missing on some parts of the pelagic food web dynamics. All major groups (along with lithogenic particles) are recorded simultaneously, which seems to indicate co-occurrence. We also observed the same patterns in the Cariaco Basin (Bringué et al. 2019, Progress in Oceanography 171: 175-211). I think part of the issue was well discussed in section 5.1, but there seems to be a decoupling between Chla and the trap record (RDA in Fig. 5 suggests that). Could it be that planktonic groups are only exported to the depths when 'ballasting minerals' are present (biogenic carbonates and silica, or lithogenics brought in by winds), but we are missing on all primary and secondary production that takes place without those denser particles? It would not undermine the data or findings, I just think it is worth discussing. Section 3.4: We usually need to justify the use of RDA by running a DCCA first (or at least a DCA on species data) – the length of the first gradient informs you on the linear vs unimodal character of the variability in the species data matrix. <2: linear and RDA is appropriate. 2-4: both ordination methods work. >4: unimodal and CCA should be used. See Canoco manual for instance. You also need to specify how the significance of the RDA ordination was tested (e.g., Monte-Carlo permutations and how permutations were done – should be the 'transect' option in Canoco because samples represent a time series; and whether the whole ordination is tested or just the first 1

or 2 axes. . .). In general, the manuscript would benefit from being revised by a native English speaker; I provide some suggestions that may help. The notation m-2d-1 should be changed throughout the manuscript and figures to include the minus signs in superscript.

Detailed comments/suggestions L. 36: Add 'are' after 'dinoflagellate cysts'. L. 66: Delete 'as' before 'together'. L. 109-111: Consider the following publications (Bringué et al. 2018, Biogeosciences 15: 1-24; Bringué et al. 2019, Progress in Oceanography 171: 175-211) that also provided multi-year records of several phytoplanktonic and (micro)zooplanktonic groups in a highly productive coastal ecosystem. L. 144: Replace 'upwelling' with 'upwelled'. L. 237: Change 'was' to 'were'. L. 263: 'saponified'. L. 293: Replace 'upheld' with 'upwelled'. L. 298: Wind directions: please specify how the variable is defined (e.g., 360° from N? or relative to coastline?) This is important to understand what the ordination in the RDA of Fig. 5 means. L. 338: Mineral dust: a reference would be desirable to support the claim, as I don't see any analysis for this stated in the methods. I don't feel strongly about this though. L. 345: Replace 'clear' with 'clearly'. L. 356: I would rather say 'variability' than 'variabilities'. L. 359: 'fluxes were'. L. 373: 'attributed to'. L. 379: 'Most noticeably'. L. 385: Rephrase to something like: '. . . whose combined contribution always represent > 50% of the community. . .' L. 389: 'G. oceanica tends to be more abundant': is this supported by flux data? I could work out that other similar statements in this paragraph are supported by the data, but this one is hard to see based on Figs. 3b and 4b. Just making sure here; but if it is, there is no need to add anything to the text. L. 399: Replace 'more distinguished' with 'more recognizable' or 'clearer'. L. 418: Replace 'are present mostly throughout' with 'are usually present throughout'. L. 436: Significance testing method needs to be detailed (in the Methods section). L. 441: Replace 'at the negative site' with 'on the negative side'. Add 'the' in 'with the exception. . .' L. 442: Add 'ordinated on' before 'positive side'. L. 461: 'Ordinated on'. L. 487: I would use 'parameters' rather than 'frames'. L. 537: Replace 'we do not disregard either its occurrence' with 'we do not dismiss its possible occurrence'. L. 547: 'into'. L. 551: Either delete 'to' in 'let us to propose' or

change to 'led us to propose'. L. 554: Replace 'yet' with 'can still be' and/or start the sentence with 'However' (and still delete 'yet'). L. 616: Change 'almost disappearance' to 'near-disappearance'. L. 621: Delete 'only' and use 'discrepancy' instead of 'lag', unless you are implying a temporal lag between temp records (or rephrase to say it is the only year with noticeable/substantial discrepencies). L. 629: Replace 'unusual' with 'unusually'. Either say 'lithogenic particles' or 'lithogenic fluxes'. L. 631: Add 'with the' after 'matches well'. L. 638: I would replace 'without adding appropriate information' with 'without providing detailed information'. L. 658: Consider replacing 'outstanding' with 'marked', 'substantial' or 'exceptional'. L. 676: Delete 'still'. Conclusions: make sure to be consistent in your use of past or present tense. L. 703: Replace 'Instead' with 'Conversely'. L. 711: Replace 'let recognizing' with 'allow (or allowed) for the recognition of'. L. 714: Make 'groups' plural. L. 719: Make sure tense is consistent. L. 723-724: Make 'patterns' plural. 'as well as'. Consider splitting that sentence in two. Figure 4: Unless the authors are planning to place this figure in one column only, the figure would greatly benefit from a legend, explaining what each color represents next to each panel. It is very difficult to refer to the caption to read the figure. Figure 5: The authors need to specify what the color-coding for labels means, as well as black vs grey arrows. I also suggest the following: - Use the abbreviation 'Chl-a' to be consistent with the text (in figure and L. 1156), - Plot organism groups as dots (circles) and variables as arrows, to more easily distinguish them (just a suggestion). Figure 6: - What is OD-phot? This group was not defined. - "C-up phot" in figure should be 'Co-up phot'. Table 1: Vertical lines are usually omitted. Table 3: Typos in 'Echinidinium spp.' and 'cyst of Protoperidinium monospinum'.

---

## Referee Comment (RC2) · Anonymous Referee #2 · 7 Oct 2019

Dr Romero and colleagues reports on the phytoplankton and zooplankton communities collected during five years by a sediment trap deployed in the Eastern Boundary Upwelling Ecosystem of the Canary Current (site CBeu in the Mauritania coastal upwelling). Authors provide detailed taxonomic information of a variety of microplankton groups including diatoms, coccolithophores, calcareous and organic dinoflagellate cysts, planktonic foraminifera and pteropods. Interestingly, authors find no evidence of succession from primary to secondary producers in their analysis. Between 2004 and 2006 authors report a change in the "normal" seasonal pattern of microplankton composition sinking into the traps mainly revealed by an increase in benthic diatom species and other changes in coccolithophore and pteropod assemblages. This signal

is interpreted by the authors as incursions of southern waters into the study region. Overall the manuscript is well-written, figures are appropriate and provides new and important insights into the functioning of upwelling ecosystems (that sustain an important fraction of global fisheries). As authors state in their manuscript, such long sediment term records of phytoplankton and zooplankton fluxes are rare but of critical importance to understand the functioning of open ocean ecosystems. This is probably the first multiannual sediment trap study in the world's ocean documenting such a large variety of microplankton groups (including both primary producers and some zooplankton groups). This type of approach is needed to improve our understanding of marine ecosystems and to establish baseline data of microplankton community composition in key regions of the global ocean. Given the value of this data set, I recommend the publication of this manuscript after some minor-moderate changes are implemented. Please find my comments below.

Title. The current title is somewhat misleading. As it reads now it seems that authors documented phyto- and zooplankton communities from the upper water column. Since phytoplankton and zooplankton assemblages can be severely altered before reaching the sediment traps I would suggest to find alternatives for the current title. Mentioning the terms sediment trap or fluxes would help to give the reader a better idea of the content of the article.

Lines 20-21. Do authors mean calcareous and organic dinoflagellate cysts? Please revise.

Lines 55-60. Authors provide a short summary of previous long-term monitoring experiments in the global ocean. As stated in the text, these type of studies are scarce, however, there are several sites in the global ocean where multiannual records of microplankton and biogeochemical fluxes have been analysed. The IOC-UNESCO report only covers very few of these sediment trap experiments. Since the current work is based on sediment trap samples, it is important to include in the introduction some of these studies in order to provide the reader with better picture of previous similar work.

Section "3.1 Moorings, sediment traps and fluxes". Can authors provide the depth of the water column at the study region? It is important to know the distance from the sea floor in order to assess the possible influence of resuspended sediments in the trap record.

Line 184: "Uncertainties with the trapping efficiency due to strong currents (e.g. under-sampling, Buesseler et al., 2007) and/or due to the migration and activity of zooplankton migrators ('swimmer problem') are assumed to be minimal in this depth range." Is this assumption based on Buesseler et al. (2007) paper? Or is this an assumption made by the authors? I would suggest to include the reference at the end of the sentence to support the whole statement and/or explain better.

Line 246-248: Authors refer to a taxonomy key that they used for dinocyst identification. However, no names of dinocyst taxa are provided in the manuscript. Why do authors provide species names for all microplankton groups but not in the case of the dinocysts? Please clarify.

Line 333: "On average, the carbonate fraction (CaCO3) dominates the mass flux (41% to the total mass flux)" Is this average a mean of all samples without considering the magnitude of the flux? Or is it an annual integrated average? I would recommend that authors provide annual values of main biogeochemical components of the flux, microplankton groups and major species in an additional figure. This information would greatly facilitate the comparison of the results of the current study with other investigations conducted in other regions of the world's ocean.

Lines 336-337. I do agree that the main contributors to CaCO3 and BSI must be the ones listed in the text. But how do authors know that the bulk of the organic carbon is delivered by diatoms coccolithophores and organic dinoflagellate cysts? Although this possibility is likely, the data provided in the current manuscript h is insufficient to reach such conclusion. In particular, in the case of diatoms, they were treated with chemicals that removed their organic content, a process that impedes the estimation of

the number of cells that reached the trap with their cellular content intact. An important fraction of the organic matter could correspond to other phytoplankton or zooplankton groups, faecal matter or other components of the marine snow. Is this statement based on previous research in the study region or is it just an interpretation by the authors? I would suggest to either provide more evidence or be more cautious with this statement. Moreover, could authors provide some insights into the contribution of the different components of the CaCO3 flux to the total CaCO3? It would be really interesting to see which microplankton groups are the most important in CaCO3 export to the deep sea.

Line 404. Why not dinocyst species are presented? Authors should provide a list of species, not only groups.

Line 416. Defining Globigerina bulloides as an upwelling species is an oversimplification of the environmental preferences of this species. The contribution of this species is often higher at times and in regions of high primary productivity, but such conditions are not necessarily linked to upwelling. Please explain better and provide references to support the affinity of this species for certain environmental conditions. Please also note that planktonic foraminifera species distribution is also influenced by changes in primary productivity not only SST as suggested in the discussion (line 582).

Line 425. "Heliconoides inflatus (formerly known as Limacina inflata)" Why two names for the same species are provided? Please clarify and provide references supporting this statement.

Lines 437-448. Authors do not mention the relationship between the different phytoplankton groups and Chlorophyll-a in Figure 5. Why? This is an important parameter that should be discussed.

Line 479. Many other studies (including sediment trap studies) have documented a simultaneous increase in the abundance of different microplankton groups during favourable conditions for phytoplankton growth. Perhaps authors could discuss their

results in light of Barber and Hiscock (2006, GBC). Please consider this suggestion and incorporate if appropriate. I found interesting that dinocysts increase their fluxes together with those of diatoms and coccolithophores. I would expect that the cysts are developed at the end of the productive period. Could authors briefly summarize/mention previous studies that describe the environmental parameters that trigger dinocyst formation?
* * *

---

## Referee Comment (RC3) · Anonymous Referee #2 · 15 Oct 2019

Just a last point to add to my review. Diatom flux data presented in this work seems to have been previously published in Romero and Fischer, 2017 (Progress in Oceanography). Could authors clarify which data is new and which data has been published before? Please note that Materials and Methods section has to be corrected accordingly. As it reads now it seems that all the diatom flux data is new. Please revise for diatoms and other microplankton groups if required.

---

## Author Comment (AC1) · 4 Nov 2019

Variability of phyto- and zooplankton communities in the Mauritanian coastal upwelling between 2003 and 2008 (bg-2019-314)

Authors = Oscar E. Romero, Karl-Heinz Baumann, Karin A. F. Zonneveld, Barbara Donner, Jens Hefter, Bambaye Hamady , Gerhard Fischer and Vera Pospelova

As required by BG, the response to the Referees is structured in the following sequence: (1) comments from Referees (RC), (2) author's response (AR), (3) author's changes in manuscript (ACM).

[Figure]

1) Comments from Dr. M. Bringué's 1.1) General comments

RC: One aspect I would like to see discussed further is the possibility of the trap record missing on some parts of the pelagic food web dynamics. All major groups (along with lithogenic particles) are recorded simultaneously, which seems to indicate co-occurrence. We also observed the same patterns in the Cariaco Basin (Bringué et al. 2019, Progress in Oceanography 171: 175-211). I think part of the issue was well discussed in section 5.1, but there seems to be a decoupling between Chla and the trap record (RDA in Fig. 5 suggests that). Could it be that planktonic groups are only exported to the depths when 'ballasting minerals' are present (biogenic carbonates and silica, or lithogenics brought in by winds), but we are missing on all primary and secondary production that takes place without those denser particles? It would not undermine the data or findings, I just think it is worth discussing.

AR: We agree with Dr. Bringué on the possible 'ballasting minerals' impact on the downward flux of primary and secondary producers. This issue was partially addressed in our first submitted version (l. 516-522), but we will extend and clarify this paragraph. This issue has been already addressed in several of our previous publication dealing with the seasonal and interannual variability of fluxes off Mauritania (e.g., Fischer et al., 2016, Biogeosciences, 13, 3071; Romero and Fischer, 2017, Prog. Oceanogr. 159, 31; Fischer et al., 2019, Global Biogeochem. Cy., 33, 1100–1128).

ACM: to be added in the revised version, Fischer et al. (2019) observed that individual high BSi maxima revealed a peak‐to‐peak correlation with winter–spring dust fluxes This was interpreted to indicate a strong coupling between dust deposition (lithogenic flux) and the efficiency of the biological pump under both dry (winter‐spring) and wet depositional conditions (summer) off Mauritania. Based on these observations, Fischer et al. (2019) proposed that the ballasted, organic‐rich aggregates built in surface waters immediately react to any additional dust supply with aggregation followed by rapid sedimentation. This was supported by experimental studies on aggregate ballasting and scavenging by v.d. Jagt et al. (2018, Limnol.

Oceanogr. 63, 1386).

RC: 2. Section 3.4: We usually need to justify the use of RDA by running a DCCA first (or at least a DCA on species data) – the length of the first gradient informs you on the linear vs unimodal character of the variability in the species data matrix. <2: linear and RDA is appropriate. 2-4: both ordination methods work. >4: unimodal and CCA should be used. See Canoco manual for instance. You also need to specify how the significance of the RDA ordination was tested (e.g., Monte-Carlo permutations and how permutations were done – should be the 'transect' option in Canoco because samples represent a time series; and whether the whole ordination is tested or just the first 1 or 2 axes...). AR & ACM: To determine if a linear or unimodal based ordination method should be applied on the data we performed a Detrended Correspondence Analysis previous to statistical analysis. Results of this analysis revealed a total length of gradient of 1.2 sd which indicates a linear species respond on environmental gradients. The methods PCA and RDA have been accordingly performed. Significance of the environmental variables have been tested with a Monte-Carlo permutation test with unrestricted permutations.

RC: In general, the manuscript would benefit from being revised by a native English speaker; I provide some suggestions that may help.

AR & ACM: Dr. Bringué's language corrections are much appreciated and will be introduced. The corrected manuscript will be reviewed by a native speaker before submission.

RF: The notation m-2d-1 should be changed throughout the manuscript and figures to include the minus signs in superscript.

AR & ACM: This will be accordingly rephrased throughout the MS. 1.2) Detailed comments/suggestions All suggested language corrections will be considered.

RC: L. 109-111: Consider the following publications (Bringué et al. 2019, Progress in

[Figure]

Oceanography 171: 175-211) that also provided multi-year records of several phyto-planktonic and (micro)zooplanktonic groups in a highly productive coastal ecosystem.

AR: this suggestion, together with Referee #2's suggestion on 'multiannual records of microplankton and biogeochemical fluxes' will be considered in the revised version. Unfortunately, there are not many long-term (longer than five years), continuous sediment trap experiments published.

ACM: in addition to Bringué et al. (2019), we will add and shortly discuss Deuser et al. (1995, Deep-Sea Res. I 42, 1923); Jickells et al. (1998, Global Biogeochem. Cy. 12, 311); and Kawahata et al. (2000, Deep-Sea Res. I 47, 2061).

RC: L. 298: Wind directions: please specify how the variable is defined (e.g., 360° from N? or relative to coastline?) This is important to understand what the ordination in the RDA of Fig. 5 means.

AR: In terms of angle measurement in degrees, 0°/360° corresponds to North, 90° to East, 180° to South and 270° to West.

ACM: This will be added to the caption of Figure 7.

RC: Figure 4: Unless the authors are planning to place this figure in one column only, the figure would greatly benefit from a legend, explaining what each color represents next to each panel. It is very difficult to refer to the caption to read the figure.

AR: this suggestion is accepted.

ACM: The names of taxa or group of taxa will be added to the right-hand side of each panel in the corrected Figure 4.

RC: Figure 5: The authors need to specify what the color-coding for labels means, as well as black vs grey arrows. I also suggest the following: - Use the abbreviation 'Chl-a' to be consistent with the text (in figure and L. 1156),

AR & ACM: we are afraid this was an unwilling problem while uploading the figure

during the submission process. We submitted an earlier version of this figure. All labels and arrows should have been in black. This will be accordingly corrected in the resubmission. Chl-a abbreviation: will be rephrased accordingly.

RC: Figure 6: "C-up phot' in figure should be 'Co-up phot'.

AR & ACM: This will be accordingly re-named.

RC: Table 1: Vertical lines are usually omitted.

AR: this suggestion is accepted.

ACM: Vertical lines in Table 1 will be omitted.

---

## Author Comment (AC2) · 4 Nov 2019

Variability of phyto- and zooplankton communities in the Mauritanian coastal upwelling between 2003 and 2008 (bg-2019-314)

Authors = Oscar E. Romero, Karl-Heinz Baumann, Karin A. F. Zonneveld, Barbara Donner, Jens Hefter, Bambaye Hamady , Gerhard Fischer and Vera Pospelova

As required by BG, the response to the Referees is structured in the following sequence: (1) comments from Referees (RC), (2) author's response (AR), (3) author's changes in manuscript (ACM).

[Figure]

Comments from Referee #2

RC: Title. The current title is somewhat misleading. As it reads now it seems that authors documented phyto- and zooplankton communities from the upper water column. Since phytoplankton and zooplankton assemblages can be severely altered before reaching the sediment traps I would suggest to find alternatives for the current title. Mentioning the terms sediment trap or fluxes would help to give the reader a better idea of the content of the article.

AR & ACM: We rephrase the MS title as: 'FLUX variability of phyto- and zooplankton communities in the Mauritanian coastal upwelling between 2003 and 2008'.

RC: Lines 20-21. Do authors mean calcareous and organic dinoflagellate cysts? Please revise.

AR & ACM: This is what we meant. It will be rephrased accordingly.

RC: Lines 55-60. Authors provide a short summary of previous long-term monitoring ex- periments in the global ocean. As stated in the text, this type of studies is scarce, however, there are several sites in the global ocean where multiannual records of microplankton and biogeochemical fluxes have been analysed. The IOC-UNESCO report only covers very few of these sediment trap experiments. Since the current work is based on sediment trap samples, it is important to include in the introduction some of these studies in order to provide the reader with better picture of previous similar work.

AR & ACM: Please see our comment above on Dr. Bringué's comments on additional references for long-term trap-based studies.

RC: Section "3.1 Moorings, sediment traps and fluxes". Can authors provide the depth of the water column at the study region? It is important to know the distance from the sea floor in order to assess the possible influence of resuspended sediments in the trap record.

AR & ACM: We will address this issue by adding an additional column in Table 1 with

the ocean bottom depth corresponding to each mooring.

RC: Line 184: "Uncertainties with the trapping efficiency due to strong currents (e.g. undersampling, Buesseler et al., 2007) and/or due to the migration and activity of zooplankton migrators ('swimmer problem') are assumed to be minimal in this depth range." Is this assumption based on Buesseler et al. (2007) paper? Or is this an assumption made by the authors? I would suggest to include the reference at the end of the sentence to support the whole statement and/or explain better.

AR & ACM: We will move Buesseler et al. (2007) reference to the end of the sentence.

RC: Line 246-248: Authors refer to a taxonomy key that they used for dinocyst identification. However, no names of dinocyst taxa are provided in the manuscript. Why do authors provide species names for all microplankton groups but not in the case of the dinocysts? Please clarify.

AR: We are afraid that Referee #2 overlooked Table 3. The originally submitted version of this Table includes all species of all groups found between June 2003 and February 2008 in samples collected with the CBeu trap.

RC: Line 333: "On average, the carbonate fraction (CaCO3) dominates the mass flux (41% to the total mass flux)" Is this average a mean of all samples without considering the magnitude of the flux? Or is it an annual integrated average? I would recommend that authors provide annual values of main biogeochemical components of the flux, mi- croplankton groups and major taxa in an additional figure. This information would greatly facilitate the comparison of the results of the current study with other investigations conducted in other regions of the world's ocean.

AR: averages depicted in Fig. 2 (horizontal dashed line for each parameter represented) and the values discussed in the MS refer to the entire sampled period (June 2003 – February 2008). We will clarify this correspondingly in the text.

ACM: In addition, we will add and discuss two new tables with annual averages of (1)

bulk components (Table 4), (2) flux of organisms, and (3) relative abundances of taxa (Table 5) represented in Figures 2-4 for those years with full calendar year sampled (2004-2007). In doing so, data will be available for future comparisons.

RC: Lines 336-337. I do agree that the main contributors to CaCO3 and BSI must be the ones listed in the text. But how do authors know that the bulk of the organic carbon is delivered by diatoms coccolithophores and organic dinoflagellate cysts? Although this possibility is likely, the data provided in the current manuscript h is insufficient to reach such conclusion. In particular, in the case of diatoms, they were treated with chemicals that removed their organic content, a process that impedes the estimation of the number of cells that reached the trap with their cellular content intact. An important fraction of the organic matter could correspond to other phytoplankton or zooplankton groups, faecal matter or other components of the marine snow. Is this statement based on previous research in the study region or is it just an interpretation by the authors? I would suggest to either provide more evidence or be more cautious with this statement. Moreover, could authors provide some insights into the contribution of the different components of the CaCO3 flux to the total CaCO3? It would be really interesting to see which microplankton groups are the most important in CaCO3 export to the deep sea.

AR: We agree with Referee #2 in that the sentence as originally written was misleading and did not express our observations properly. This will be accordingly re-phrased. Since we did not perform any quantitative analysis on how much organic matter each studied plankton group is contributing with, we are not able to provide any absolute or relative values of contribution of particular group/s to either organic carbon and/or calcium carbonate. Part of our current work on the seasonal and interannual variability of microorganism and bulk fluxes off Mauritania focuses on the issue of how much each of the calcareous groups contribute to the flux of CaCO3 as measured in trap samples. Although we are not able to provide any actual values at this stage, we are still convinced that diatoms and coccolithophores are important contributors to the flux

of organic carbon at the CBeu site.

ACM: We will add following sentence, Diatoms, coccolithophores and organic dinoflagellates are important contributors to the flux of organic carbon at the CBeu site.

RC: Line 404. Why not dinocyst species are presented? Authors should provide a list of species, not only groups.

AR: as commented above, the originally submitted version of this Table includes all species of all groups found between June 2003 and February 2008 in samples collected with the CBeu trap.

RC: Line 416. Defining Globigerina bulloides as an upwelling species is an oversimplification of the environmental preferences of this species. The contribution of this species is often higher at times and in regions of high primary productivity, but such conditions are not necessarily linked to upwelling. Please explain better and provide references to support the affinity of this species for certain environmental conditions. Please also note that planktonic foraminifera species distribution is also influenced by changes in primary productivity not only SST as suggested in the discussion (line 582).

AR: we revised the ecological interpretation of G. bulloides' temporal occurrence at site CBeu. The planktonic foraminifera group including G. bulloides will be re-named 'high nutrient waters' (instead of 'upwelling').

ACM: We will rephrase the corresponding sentences as follows: Globigerina bulloides is usually associated with temperate to sub-polar water masses and seasonally enhanced primary production due to increased nutrient availability. Additionally, G. bulloides is a common fauna component in low-latitude areas influenced by upwelling (compilation in Schiebel and Hemleben, 2017, Planktic Foraminifers in the Modern Ocean). At site CBeu, it is generally more abundant between summer and fall (Fig. 4).

RC: Line 425. "Heliconoides inflatus (formerly known as Limacina inflata)" Why two names for the same species are provided? Please clarify and provide references supporting this statement.

AR & ACM: this will be properly rephrased and only the name 'Heliconoides inflatus" will be used throughout the MS and Figures. However, we will present the former name in brackets) in Table 3.

RC: Lines 437-448. Authors do not mention the relationship between the different phytoplankton groups and Chlorophyll-a in Figure 5. Why? This is an important parameter that should be discussed.

AR: we do agree with both Referees in that this should have been discussed in the first submitted version. A short discussion will be added in the revised version.

ACM: Interestingly, the satellite-gained data we used for our statistical analysis do not show a significant correlation with fluxes of major microorganism groups studied at site CBeu. This possibly indicates that (1) a large portion of satellite-measured chlorophyll concentration is delivered by microorganisms, which do not reach the CBeu trap, and/or (2) due to strong ballasting effect, part of the microorganisms' remains reach the trap cups independent of intervals of highest chlorophyll values as measured by satellites. An alternative explanation is the fact that (3) satellites measure chlorophyll concentration in the uppermost centimeters of the water column while those microorganisms collected with the CBeu traps thrive mostly in waters deeper than those reached by satellite sensors. All three explanations for this observation will be discussed in the revised version.

RC: Line 479. Many other studies (including sediment trap studies) have documented a simultaneous increase in the abundance of different microplankton groups during favorable conditions for phytoplankton growth. Perhaps authors could discuss their results in light of Barber and Hiscock (2006, GBC). Please consider this suggestion and incorporate if appropriate. I found interesting that dinocysts increase their fluxes together with those of diatoms and coccolithophores. I would expect that the cysts are developed at the end of the productive period. Could authors briefly summarize/mention previous

studies that describe the environmental parameters that trigger dinocyst formation?

AR: Barber and Hiscock (2006, Global Biogeochem. Cy 20, GB4S03) will be discussed in the revised version.

ACM: Barber and Hiscock (2006) observed that marine picoplankton is not replaced by diatoms when chemical transient (e.g., added iron) abruptly provided favorable growth conditions. Contrary to conventional wisdom, both groups of phytoplankton increase in growth rates and absolute abundance, but the biomass increase of the ambient non-diatom assemblage is modest, especially compared to the order of magnitude or more increase of diatom biomass. This enormous proportional increase in diatom biomass has fostered the misconception that diatoms replace the non-diatom taxa by succession as the bloom matures.

RC: I found interesting that dinocysts increase their fluxes together with those of diatoms and coccolithophores. I would expect that the cysts are developed at the end of the productive period. Could authors briefly summarize/mention previous studies that describe the environmental parameters that trigger dinocyst formation?

AR: This suggestion is accepted and will be accordingly addressed in the revised version.

ACM: Sexuality and the formation of gametes of dinoflagellates can be triggered by a number of environmental factors such as nutrient limitation, darkness, suboptimal temperature, endogenic rhythms, cell density and day length (e.g., Anderson et al., 1985; Ellegaard et al., 1998; Sgrosso et al., 2001; Uchida, 2001; Figueroa and Bravo, 2005; Kremp et al., 2009). In field studies, sexual reproduction generally starts at periods with high vegetative cell concentrations in the water column and maximal cyst production occurs notably at the termination of dinoflagellate blooms (e.g., Matsuoka and Takeuchi, 1995; Bravo, et al., 2010; Figueroa et al., 2018). With exception of the calcareous dinoflagellate cysts Leonella granifera and Thoracosphaera heimii that are known to be produced during the vegetative life cycle, the calcareous and organicwalled cysts of all other dinoflagellate species can be considered to be produced as part of their sexual life cycle (e.g., Meier et al., 2007; Bravo and Figueroa, 2014, and references therein). As a consequence, high numbers of trap-collected cysts can be considered to be the result of high abundance of motile cells of these species in the upper water column.

---

## Author Comment (AC3) · 4 Nov 2019

Variability of phyto- and zooplankton communities in the Mauritanian coastal upwelling between 2003 and 2008 (bg-2019-314)

Authors = Oscar E. Romero, Karl-Heinz Baumann, Karin A. F. Zonneveld, Barbara Donner, Jens Hefter, Bambaye Hamady , Gerhard Fischer and Vera Pospelova

As required by BG, the response to the Referees is structured in the following sequence: (1) comments from Referees (RC), (2) author's response (AR), (3) author's changes in manuscript (ACM).

Comments from Referee #2

RC: Just a last point to add to my review. Diatom flux data presented in this work seems to have been previously published (Romero and Fischer, 2017, Prog. Oceanogr. 159, 31-44). Could authors clarify which data is new and which data has been published before? Please note that Materials and Methods section has to be corrected accordingly. As it reads now it seems that all the diatom flux data is new. Please revise for diatoms and other microplankton groups if required.

AR: All coccolithophore, dinoflagellate, planktonic foraminifera and pteropod data are new. Diatom data were previously published in Romero and Fischer (2019, Prog. Ocean 159, 31). We will make clear that our MS builds on previous work.

ACM: In addition to the previously published diatom data (Romero and Fischer, 2017), we present here new data on fluxes of coccolithophores, calcareous and organic dinoflagellate cysts, planktonic foraminifera and pteropods, and the relative contribution (percentage) of most ecologically important taxa collected with trap CBeu between June 2003 and February 2008.

---

## Author Response (AR1)

Tel. +49 421 218 – 65 645 E-Mail oromero@uni-bremen.de www www.marum.de

November 18th 2019

*Biogeosciences* Associate Editor Prof. Emilio Marañón Universidad de Vigo Spain

**Dear Emilio,**

We submit the revised version of our manuscript "*Flux variability of phyto-and zooplankton communities in the Mauritanian coastal upwelling between 2003 and 2008*" (bg-2019-314). We have endeavored to deal with all of the issues raised by both referees. Following both reviews, several changes were made to the text, figures and tables. In addition to the point-by-point response to the remarks raised by Dr. Manuel Bringué, an anonymous referee and yourself, we have included following changes:

- (1) The title has been rephrased.
- (2) A new co-author has been added (Prof. Vera Pospelova).
- (3) Two new tables have been added (now Tables 3 and 4).

In addition to our Reply and the point-by-point reply to the comments, the following documents are submitted:

- (1) a copy of the revised manuscript (the marked-up version shows all changes made throughout the text as highlighted in red), and,
- (2) all files in publication-ready formats (including high-resolution files of figures).

We greatly appreciate the helpful reviewers' insights. We hope that this revised version will merit your positive consideration and the editorial requirements of *Biogeosciences*.

Best regards,

Jun

Oscar E. Romero (on behalf of all co-authors)

**Flux variability of phyto- and zooplankton communities in the Mauritanian coastal upwelling between 2003 and 2008**

**bg-2019-314**

Authors = Oscar E. Romero, Karl-Heinz Baumann, Karin A. F. Zonneveld, Barbara Donner, Jens Hefter, Bambaye Hamady, Vera Pospelova and Gerhard Fischer

**Reply to the Interactive Discussion**

As required by BG, the response to the Referees is structured in the following sequence: (1) comments from Referees (RC), (2) author's response (AR), (3) author's changes in manuscript (ACM).

**1) Comments from Dr. M. Bringué's**

**1.1) General comments**

RC: One aspect I would like to see discussed further is the possibility of the trap record missing on some parts of the pelagic food web dynamics. All major groups (along with lithogenic particles) are recorded simultaneously, which seems to indicate co-occurrence. We also observed the same patterns in the Cariaco Basin (Bringué et al. 2019, Progress in Oceanography 171: 175-211). I think part of the issue was well discussed in section 5.1, but there seems to be a decoupling between Chla and the trap record (RDA in Fig. 5 suggests that). Could it be that planktonic groups are only exported to the depths when 'ballasting minerals' are present (biogenic carbonates and silica, or lithogenics brought in by winds), but we are missing on all primary and secondary production that takes place without those denser particles? It would not undermine the data or findings, I just think it is worth discussing.

AR: We agree with Dr. Bringué on the possible 'ballasting minerals' impact on the downward flux of primary and secondary producers. This issue was partially addressed in our first submitted version (l. 516-522), but we will extend and clarify this paragraph. This issue has been already addressed in several of our previous publication dealing with the seasonal and interannual variability of fluxes off Mauritania (e.g., Fischer et al., 2016, Biogeosciences, 13, 3071; Romero and Fischer, 2017, Prog. Oceanogr. 159, 31; Fischer et al., 2019, Global Biogeochem. Cy., 33, 1100–1128).

ACM: added to the revised version, Further evidence of a possible ballasting effect on the flux of bulk components was recently presented by Fischer et al. (2019). These authors observed that individual high BSi maxima at site CBeu revealed a peak-to-peak correlation with the dust fluxes. This was interpreted to indicate a strong coupling between dust deposition (lithogenic flux) and the efficiency of the biological pump under dry depositional conditions in winter off Mauritania. Based on these observations, Fischer et al. (2019) proposed that the ballasted, organic-rich aggregates built in surface waters immediately react to any additional dust supply with aggregation followed by rapid sedimentation. Experimental studies on aggregate ballasting and scavenging off

*Mauritania (van der Jagt et al., 2018) support this view as well.* (l. 550-558 in revised version).

RC: 2. Section 3.4: We usually need to justify the use of RDA by running a DCCA first (or at least a DCA on species data) – the length of the first gradient informs you on the linear vs unimodal character of the variability in the species data matrix. <2: linear and RDA is appropriate. 2-4: both ordination methods work. >4: unimodal and CCA should be used. See Canoco manual for instance. You also need to specify how the significance of the RDA ordination was tested (e.g., Monte-Carlo permutations and how permutations were done – should be the 'transect' option in Canoco because samples represent a time series; and whether the whole ordination is tested or just the first 1 or 2 axes...).

AR & ACM: To determine if a linear or unimodal based ordination method should be applied on the data we performed a Detrended Correspondence Analysis previous to statistical analysis. Results of this analysis revealed a total length of gradient of 1.2 sd which indicates a linear species respond on environmental gradients. The methods PCA and RDA have been accordingly performed. Significance of the environmental variables have been tested with a Monte-Carlo permutation test with unrestricted permutations. (l. 315-320 in revised version).

RC: In general, the manuscript would benefit from being revised by a native English speaker; I provide some suggestions that may help.

AR & ACM: Dr. Bringué's language corrections are much appreciated and will be introduced. The corrected manuscript was reviewed by Prof. Vera Pospelova (Canada).

RF: The notation m-2d-1 should be changed throughout the manuscript and figures to include the minus signs in superscript.

AR & ACM: This was accordingly rephrased throughout the MS.

**1.2)** Detailed comments/suggestions**

All suggested language corrections will be considered.

RC: L. 109-111: Consider the following publications (Bringué et al. 2019, Progress in Oceanography 171: 175-211) that also provided multi-year records of several phytoplanktonic and (micro)zooplanktonic groups in a highly productive coastal ecosystem.

AR: this suggestion, together with Referee #2's suggestion on 'multiannual records of microplankton and biogeochemical fluxes' was considered in the revised version. Unfortunately, there are not many long-term (longer than five years), continuous sediment trap experiments dealing with several groups of primary and secondary producers.

ACM: we mentioned and shortly discuss Bringué et al. (2019.

RC: L. 298: Wind directions: please specify how the variable is defined (e.g., 360° from N? or relative to coastline?) This is important to understand what the ordination in the RDA of Fig. 5 means.

AR: In terms of angle measurement in degrees,  $0^{\circ}/360^{\circ}$  corresponds to North,  $90^{\circ}$  to East,  $180^{\circ}$  to South and  $270^{\circ}$  to West.

ACM: This was rephrased in Figure 7.

RC: Figure 4: Unless the authors are planning to place this figure in one column only, the figure would greatly benefit from a legend, explaining what each color represents next to each panel. It is very difficult to refer to the caption to read the figure.

AR: this suggestion is accepted.

ACM: The names of taxa or group of taxa was added to the right-hand side of each panel in the corrected Figure 4.

RC: Figure 5: The authors need to specify what the color-coding for labels means, as well as black vs grey arrows. I also suggest the following:

- Use the abbreviation 'Chl-a' to be consistent with the text (in figure and L. 1156), AR & ACM: we are afraid this was an unwilling problem while uploading the figure during the submission process. We submitted an earlier version of this figure. All labels and arrows should have been in black. This is now corrected.

RC: Figure 6: "C-up phot' in figure should be 'Co-up phot'. AR & ACM: *This has been accordingly re-named.*

RC: Table 1: Vertical lines are usually omitted.

AR: this suggestion is accepted.

ACM: Vertical lines in Table 1 are now omitted.

**Flux variability of phyto- and zooplankton communities in the Mauritanian coastal upwelling between 2003 and 2008 (bg-2019-314)**

Authors = Oscar E. Romero, Karl-Heinz Baumann, Karin A. F. Zonneveld, Barbara Donner, Jens Hefter, Bambaye Hamady, Vera Pospelova and Gerhard Fischer

As required by BG, the response to the Referees is structured in the following sequence: (1) comments from Referees (RC), (2) author's response (AR), (3) author's changes in manuscript (ACM).

**Comments from Referee #2**

RC: Title. The current title is somewhat misleading. As it reads now it seems that authors documented phyto- and zooplankton communities from the upper water column. Since phytoplankton and zooplankton assemblages can be severely altered before reaching the sediment traps I would suggest to find alternatives for the current title. Mentioning the terms sediment trap or fluxes would help to give the reader a better idea of the content of the article.

AR & ACM: We rephrase the MS title as: 'FLUX variability of phyto- and zooplankton communities in the Mauritanian coastal upwelling between 2003 and 2008'.

RC: Lines 20-21. Do authors mean calcareous and organic dinoflagellate cysts? Please revise.

AR & ACM: This was accordingly rephrased.

RC: Lines 55-60. Authors provide a short summary of previous long-term monitoring experiments in the global ocean. As stated in the text, this type of studies is scarce, however, there are several sites in the global ocean where multiannual records of microplankton and biogeochemical fluxes have been analysed. The IOC-UNESCO report only covers very few of these sediment trap experiments. Since the current work is based on sediment trap samples, it is important to include in the introduction some of these studies in order to provide the reader with better picture of previous similar work.

AR & ACM: Please see our comment above on Dr. Bringué's comments on additional references for long-term trap-based studies.

RC: Section "3.1 Moorings, sediment traps and fluxes". Can authors provide the depth of the water column at the study region? It is important to know the distance from the sea floor in order to assess the possible influence of resuspended sediments in the trap record.

AR & ACM: We addressed this by adding an additional column in Table 1 with the ocean bottom depth corresponding to each mooring.

RC: Line 184: "Uncertainties with the trapping efficiency due to strong currents (e.g. undersampling, Buesseler et al., 2007) and/or due to the migration and activity of zooplankton migrators ('swimmer problem') are assumed to be minimal in this depth range." Is this assumption based on Buesseler et al. (2007) paper? Or is this an assumption made by the authors? I would suggest to include the reference at the end of the sentence to support the whole statement and/or explain better.

AR & ACM: Buesseler et al. (2007) was moved to the end of the sentence.

RC: Line 246-248: Authors refer to a taxonomy key that they used for dinocyst identification. However, no names of dinocyst taxa are provided in the manuscript. Why do authors provide species names for all microplankton groups but not in the case of the dinocysts? Please clarify.

AR: We are afraid that Referee #2 overlooked Table 3 (now Table 5). The originally submitted version of this Table includes all species of all groups found between June 2003 and February 2008 in samples collected with the CBeu trap.

RC: Line 333: "On average, the carbonate fraction (CaCO3) dominates the mass flux (41% to the total mass flux)" Is this average a mean of all samples without considering the magnitude of the flux? Or is it an annual integrated average? I would recommend that authors provide annual values of main biogeochemical components of the flux, microplankton groups and major taxa in an additional figure. This information would greatly facilitate the comparison of the results of the current study with other investigations conducted in other regions of the world's ocean.

*AR: averages depicted in Fig. 2 (horizontal dashed line for each parameter represented) and the values discussed in the MS refer to the entire sampled period (June 2003 – February 2008).*

ACM: We added two new tables with annual averages of (1) bulk components (Table 3), (2) flux of organisms, and (3) relative abundances of taxa (Table 4) represented in Figures 2-4 for those years with full calendar year sampled (2004-2007). In doing so, data will be available for future comparisons.

RC: Lines 336-337. I do agree that the main contributors to CaCO3 and BSI must be the ones listed in the text. But how do authors know that the bulk of the organic carbon is delivered by diatoms coccolithophores and organic dinoflagellate cysts? Although this possibility is likely, the data provided in the current manuscript h is insufficient to reach such conclusion. In particular, in the case of diatoms, they were treated with chemicals that removed their organic content, a process that impedes the estimation of the number of cells that reached the trap with their cellular content intact. An important fraction of the organic matter could correspond to other phytoplankton or zooplankton

groups, faecal matter or other components of the marine snow. Is this statement based on previous research in the study region or is it just an interpretation by the authors? I would suggest to either provide more evidence or be more cautious with this statement. Moreover, could authors provide some insights into the contribution of the different components of the CaCO3 flux to the total CaCO3? It would be really interesting to see which microplankton groups are the most important in CaCO3 export to the deep sea.

AR: We agree with Referee #2 in that the sentence as originally written was misleading and did not express our observations properly. This was accordingly rephrased.

ACM: We added the following sentence, *Coccolithophores, planktonic foraminifera, calcareous dinoflagellates and pteropods are main contributors to the CaCO*3 *flux, while diatoms dominate the siliceous community.* (l. 356-58 in revised version)

RC: Line 404. Why not dinocyst species are presented? Authors should provide a list of species, not only groups.

AR: as commented above, the originally submitted version of Table 3 (now T 5) includes all species of all groups found between June 2003 and February 2008 in samples collected with the CBeu trap.

RC: Line 416. Defining Globigerina bulloides as an upwelling species is an oversimplification of the environmental preferences of this species. The contribution of this species is often higher at times and in regions of high primary productivity, but such conditions are not necessarily linked to upwelling. Please explain better and provide references to support the affinity of this species for certain environmental conditions. Please also note that planktonic foraminifera species distribution is also influenced by changes in primary productivity not only SST as suggested in the discussion (line 582).

AR: we revised the ecological interpretation of G. bulloides' temporal occurrence at site CBeu. The planktonic foraminifera group including G. bulloides will be re-named 'high nutrient waters' (instead of 'upwelling').

ACM: We rephrased the corresponding sentences as follows: Globigerina bulloides, usually thriving in temperate to subpolar waters during intervals of enhanced primary productivity due to high nutrient availability (Schiebel and Hemleben, 2017), is generally most abundant between summer and fall (fig. 4e) (l. 435-38 in revised version).

RC: Line 425. *"Heliconoides inflatus* (formerly known as *Limacina inflata*)" Why two names for the same species are provided? Please clarify and provide references supporting this statement.

AR & ACM: this will be properly rephrased and only the name 'Heliconoides inflatus" will be used throughout the MS and Figures. The former name is only given in Table 3.

RC: Lines 437-448. Authors do not mention the relationship between the different phytoplankton groups and Chlorophyll-a in Figure 5. Why? This is an important parameter that should be discussed.

AR: we do agree with both Referees in that this should have been discussed in the first submitted version.

ACM: following paragraph has been added, *Interestingly, the satellite-derived Chl-a data* do not show a significant correlation with fluxes of major microorganism groups studied at site CBeu. This possibly indicates that (1) a large portion of satellite-measured chlorophyll concentration is delivered by microorganisms, which did not reach the CBeu trap, and/or (2) due to the strong ballasting effect, part of the microorganisms' remains reach the trap cups independent of intervals of highest satellite chlorophyll values. An alternative explanation is that (3) satellites measure the chlorophyll concentration in the uppermost centimeters of the water column while microorganisms collected with the CBeu traps thrive mostly in waters deeper than those reached by satellite sensors. (l. 469-476 in revised version)

RC: Line 479. Many other studies (including sediment trap studies) have documented a simultaneous increase in the abundance of different microplankton groups during favorable conditions for phytoplankton growth. Perhaps authors could discuss their results in light of Barber and Hiscock (2006, GBC). Please consider this suggestion and incorporate if appropriate. I found interesting that dinocysts increase their fluxes together with those of diatoms and coccolithophores. I would expect that the cysts are developed at the end of the productive period. Could authors briefly summarize/mention previous studies that describe the environmental parameters that trigger dinocyst formation?

AR: Barber and Hiscock (2006) and others are now discussed in the revised version.

ACM: Following paragraph was added, Although it is widely believed that the supply of resources regulates the marine community structure (Roelke and Spatharis, 2015), experimental data show that the competition of resources per se does not lead to succession of phytoplankton populations. For instance, Barber and Hiscock (2006) observed that marine picoplankton is not replaced by diatoms when chemical transient conditions (e.g., added iron) abruptly provide a more favorable growth setting. The enormous proportional increase in diatom biomass has fostered the misconception that diatoms replace the non-diatom taxa by succession as the bloom matures. Additional evidence is provided by observational studies. Bringué et al. (2018) observed that autotrophic dinoflagellates in the Cariaco Basin do not appear to compete with diatoms for resources as both groups respond positively to upwelling dynamics. Similarly, Anabalón et al. (2014) observed equal and simultaneous contributions of diatoms and pigmented dinoflagellates to total autotrophic biomass off Cape Ghir located in the northern CC-EBUEs. Contrary to conventional wisdom, Anabalón et al. (2014) demonstrate that groups of phytoplankton increase in growth rates and absolute abundance, but the biomass

increase of the ambient non-diatom assemblage is modest, especially compared to the order of magnitude or more increase of diatom biomass. (l. 569-583 in revised version).

**Flux variability of phyto- and zooplankton communities in the Mauritanian coastal upwelling between 2003 and 2008 (bg-2019-314)**

Authors = Oscar E. Romero, Karl-Heinz Baumann, Karin A. F. Zonneveld, Barbara Donner, Jens Hefter, Bambaye Hamady, Vera Pospelova and Gerhard Fischer

As required by BG, the response to the Referees is structured in the following sequence: (1) comments from Referees (RC), (2) author's response (AR), (3) author's changes in manuscript (ACM).

**Comments from Referee #2**

RC: Just a last point to add to my review. Diatom flux data presented in this work seems to have been previously published (Romero and Fischer, 2017, Prog. Oceanogr. 159, 31-44). Could authors clarify which data is new and which data has been published before? Please note that Materials and Methods section has to be corrected accordingly. As it reads now it seems that all the diatom flux data is new. Please revise for diatoms and other microplankton groups if required.

AR: All coccolithophore, dinoflagellate, planktonic foraminifera and pteropod data are new. Diatom data were previously published in Romero and Fischer (2019, Prog. Ocean 159, 31). We will make clear that our MS builds on previous work.

ACM: Detailed information about sampling and laboratory analysis is given in Mollenhauer et al. (2015), Romero and Fischer (2017) and Fischer et al. (2019). These papers present the bulk fluxes for the deployments CBeu 1-5 (Table 1). Alkenone-derived sea surface temperature (SST) for the CBeu deployments 1-4 were provided by Mollenhauer et al. (2015). (l. 202-206 of the revised version).

**Flux variability of phyto- and zooplankton communities**

2

Oscar E. Romero1, Karl-Heinz Baumann1,2, Karin A. F. Zonneveld1, Barbara Donner1, Jens Hefter3,
 Bambaye Hamady4, Vera Pospelova5,6 and Gerhard Fischer1,2

[revised manuscript text omitted]